# DASFormer: Self-supervised Pretraining for Earthquake Monitoring

## Abstract

Earthquake monitoring is a fundamental task to unravel the underlying physics of earthquakes and mitigate associated hazards for public safety. Distributed acoustic sensing, or DAS, which transforms pre-existing telecommunication cables into ultra-dense seismic networks, offers a cost-effective and scalable solution for next-generation earthquake monitoring. However, current approaches for earthquake monitoring primarily rely on supervised learning, while manually labeled DAS data is quite limited and it is difficult to obtain more annotated datasets. In this paper, we present DASFormer, a novel self-supervised pretraining technique on DAS data with a coarse-to-fine framework that models spatial-temporal signal correlation. We treat earthquake monitoring as an anomaly detection task and demonstrate DASFormer can be directly utilized as a seismic phase detector. Experimental results demonstrate that DASFormer is effective in terms of several evaluation metrics and outperforms state-of-the-art time-series forecasting, anomaly detection, and foundation models on the unsupervised seismic detection task. We also demonstrate the potential of fine-tuning DASFormer to downstream tasks through case studies.

## 1 Introduction

Earthquake, as a natural disaster, poses a constant threat to public safety due to its randomness and potentially catastrophic damage. Monitoring earthquakes using large networks of seismic sensors, such as seismometers, is a fundamental task to unravel the underlying physics and mitigate associated hazards Ringler et al. (2022). Conventional seismic networks with sensor spacing of 10 to 100 km limit the capability of earthquake monitoring at finer scales. To overcome the bottleneck, a new technology called Distributed Acoustic Sensing (DAS) offers a cost-effective and scalable solution Hartog (2017); Zhan (2020); Lindsey & Martin (2021). Figure 1 illustrates how DAS works: by sensing ground motion from back-scattered laser light due to fiber defects, DAS can transform ubiquitous telecommunication cables into ultra-dense monitoring networks. With thousands of seismic sensors in meter-scale spacing, DAS continuously records a wide range of natural signals, paving the way for next-generation earthquake monitoring Zhan (2020); Lindsey & Martin (2021).

Recently, deep learning techniques have achieved progressive breakthroughs in extensive areas. Deep-learning-based methods Ross et al. (2018); van den Ende et al. (2021b); Zhu & Beroza (2019); Spoorthi et al. (2020); Kuang et al. (2021); Smith et al. (2020); Dahmen et al. (2022); Mousavi et al. (2020); McBrearty & Beroza (2023); Sun et al. (2023); Smith et al. (2022) have been credited with significant advancements in earthquake monitoring. However, earthquake monitoring using DAS remains challenging due to the lack of manually labeled DAS data, especially P and S phase labels that carry vital information about seismic sources. Furthermore, it is a fundamental problem to generalize deep foundation models well for a wide range of downstream tasks, such as denoising, P/S phase picking, etc. Current studies generally build on labeled or synthetic datasets Ross et al. (2018); Zhu & Beroza (2019); Spoorthi et al. (2020); Kuang et al. (2021); Smith et al. (2020); Dahmen et al. (2022); Mousavi et al. (2020); McBrearty & Beroza (2023); Sun et al. (2023); Smith et al. (2022) or focus on denoising task van den Ende et al. (2021b).

Inspired by the success of BERT Devlin et al. (2019) in natural language processing, we develop a novel self-supervised framework named DASFormer, which is pre-trained on a mask prediction task to empower the representation ability for DAS data. For the ability for downstream tasks,

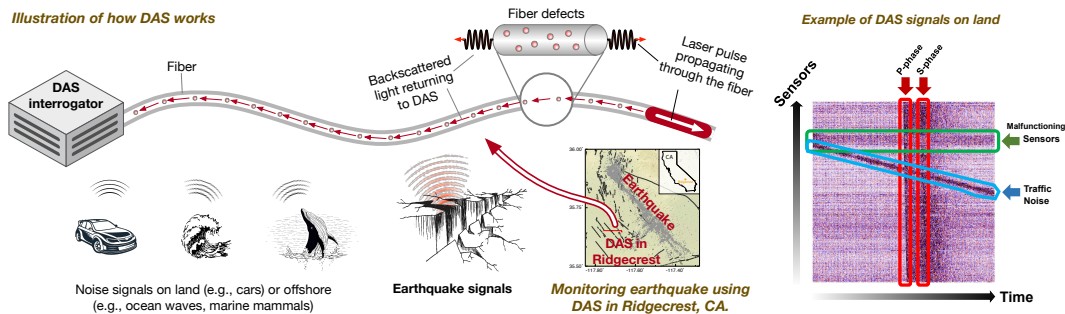

Figure 1: Illustration of how Distributed Acoustic Sensing (DAS) works for earthquake monitoring. An example of DAS data collected in Ridgecrest City, CA is shown on the right.

we demonstrate that the well-pretrained DASFormer can be served as a time series detector for earthquake monitoring. The underlying assumption here is that earthquakes occur randomly and thus are much more challenging to causally predict alongside the time direction than other sources. Figure 1 shows an example that earthquake-generated P and S phases sharply appear as vertical anomalies on the DAS data whereas malfunctioning sensors and traffic signals have more temporal causality. Furthermore, we also demonstrate the potential of DASFormer on other downstream tasks like malfunctioning detection, semi-supervised P/S phase picking prediction, etc. In our practical experiments, our method outperforms a variety of state-of-the-art time series forecasting, anomaly detection, and time series foundation models on realistic DAS data collected in Ridgecrest city, CA Li et al. (2021a).

The contributions of this paper can be summarized as follows:

- We propose a novel self-supervised framework for DAS data named DASFormer, which can be applied to downstream tasks for earthquake monitoring. DASFormer leverages the unique characteristics of DAS data to learn meaningful representations without the need for manually labeled data. By exploiting the temporal and spatial information captured by DAS, our framework can effectively capture the dynamic nature of seismic activities and extract valuable features for earthquake monitoring.

- The model comes with a well-designed coarse-to-fine framework built on the top of Swin U-Net Cao et al. (2022) and Convolutional U-Net Ronneberger et al. (2015) to effectively capture both spatial and temporal patterns in DAS data. To further enhance the framework, we introduce several key components such as preprocessing strategy, patching strategy, DASFormer blocks, and GAN-like training scheme Goodfellow et al. (2020).

- We demonstrate the effectiveness of our method mainly on the task of unsupervised P/S phase detection on realistic DAS data. Our experimental results show that DASFormer outperforms 13 and 7 state-of-the-art methods for time series forecasting and anomaly detection, respectively, as well as 2 advanced time series foundation models. Furthermore, we also show the potential for other downstream tasks through case studies.

## 2 RELATED WORK

**Deep Learning on DAS Data** Treating DAS data as time series with deep learning (DL) methods enhances sensitivity to small-magnitude earthquake signals compared to traditional techniques like STA/LTA Allen (1978) or template matching Gibbons & Ringdal (2006). Notwithstanding, lacking precise arrival times for P/S phases remains a crucial challenge for the aforementioned methods. New deep-learning-based methods such as PhaseNet Zhu & Beroza (2019) and PhaseNet-DAS Zhu et al. (2023) used supervised U-Net networks for P/S phase picking when large datasets with manual labels are available; FMNet Kuang et al. (2021), EikoNet Smith et al. (2020), and MarsQuakeNet Dahmen et al. (2022) apply synthetic data to bypass label scarcity; $j$-DAS van den Ende et al. (2021a) offers a self-supervised framework only for signal denoising. This paper presents DASFormer, a self-supervised technique effective in both unlabeled and minimally labeled scenarios, with the potential for various downstream tasks.

**Time Series Modeling** Temporal variation modeling, integral to time series analysis, has seen significant advancements recently. Classical methods such as the ARIMA families Box et al. (2015) were proposed based on prior assumptions of temporal variations. However, these fail to capture the non-stationarity and complexity of real-world time series. Consequently, deep learning-based models for time series have emerged, adapting MLPs Oreshkin et al. (2020); Challu et al. (2022); Zeng et al. (2022); Zhang et al. (2022) to time-series data and introducing temporal-aware neural networks like TCN Franceschi et al. (2019), RNN and LSTM Hochreiter & Schmidhuber (1997a). Recently, Transformer Vaswani et al. (2017) has garnered attention for time series modeling, with architectures like Informer Zhou et al. (2021), LogTrans Li et al. (2019b), and ASTrans Wu et al. (2020) addressing resource consumption issues. Autoformer Wu et al. (2021) and FEDformer Zhou et al. (2022b) focus on novel decomposition schemes for trend-cyclical component extraction in time series. TimesNet Wu et al. (2022) proposes a task-general foundation model for time series analysis. Moreover, PatchTST Nie et al. (2023) and ViTST Li et al. (2023) adapt the Vision Transformer (ViT) Dosovitskiy et al. (2021) for time series modeling, akin to our method. However, both of them use time series line graphs to illustrate temporal data points, while our method directly utilizes the magnitude of time points as values, leveraging the scalable window mechanism in Swin U-Net Liu et al. (2021b); Cao et al. (2022); Li et al. (2022) to learn spatial and temporal patterns. Unlike previous methods that overlook the correlation and invariance in multi-variate time series like DAS data, our model considers spatial patterns in variates to complement temporal patterns, serving as a general foundation model.

**Anomaly Detection** The P/S-phase detection task is a primary task in earthquake monitoring, which can be viewed as an anomaly detection problem. Anomaly detection methods typically fall into three categories: clustering-based, reconstruction-based, and forecasting-based methods. Clustering-based methods Tax & Duin (2004); Ruff et al. (2018a) measure anomaly scores based on the sample's distance to the cluster center. Reconstruction-based methods evaluate anomaly scores via the reconstruction error. Deep learning-based generative models like Variational AutoEncoders (VAEs) Park et al. (2018); Su et al. (2019); Li et al. (2021b) and GANs Goodfellow et al. (2014); Li et al. (2019a); Schlegl et al. (2019); Zhou et al. (2019) have been widely investigated to generate reconstructed time series for anomaly detection. Recently, Anomaly Transformer Xu et al. (2022) proposes to utilize the great sequential modeling capacity of Transformer Vaswani et al. (2017) and renovate the self-attention mechanism specific to anomaly detection tasks. Forecasting-based methods Hundman et al. (2018); Tariq et al. (2019) typically leverage temporal models like ARIMA and LSTMs to forecast time series, identifying anomalies based on the discrepancy between predicted and actual values. This paper illustrates how our base model can be used directly as a forecasting-based anomaly detection method for the P/S phase detection task in earthquake monitoring.

## 3 PROBLEM FORMULATION OF EARTHQUAKE DETECTION

In this paper, our main downstream task is the earthquake detection, as known as P/S phase detection, which is based on the DAS data, a kind of spatial-temporal multi-variate time series data. Formally, assume we have $N$ DAS sensors that record measurements over a period of $T$ time units. Let $\mathbf{X} \in \mathbb{R}^{N \times T}$ denote the DAS collected data matrix, with $\mathbf{X}_{i,j}$ being the measurement recorded by the $i$-th sensor at time $j$. The task of earthquake detection is to identify earthquake signals from background noise and other signals in the data matrix $\mathbf{X}$. The labels categorize each measurement $\mathbf{X}_{i,j}$ into binary $1/0$ indicating whether an earthquake signal is present.

As a new technology, DAS has limited annotated earthquake labels. We tackle the lack of ground truth labels by considering an unsupervised setting in this paper. We train our model on a massive number of unlabeled raw DAS data, and evaluate it on a small labeled DAS signal subset. To effectively capture the temporal dynamics of DAS data, we pose this problem as a multi-variate time series forecasting task, which can be formulated as follows:

Given a multi-variate time series of DAS data from multiple sensors over a historical period $k$ observed at the current time $t$, $\mathbf{I} = \{X_{:,t-k}, X_{:,t-k+1}, ..., X_{:,t}\}$, the goal is to train a model to predict the future time series in $p$ time steps $\mathbf{O} = \{X_{:,t+1}, X_{:,t+2}, ..., X_{:,t+p}\}$. When an earthquake suddenly occurs, it generates seismic waves (i.e., P and S wave) that vibrate the fiber cables and are recorded by DAS. Because the occurrence of earthquakes is temporally unpredictable, the difference between predicted and real data becomes larger, implying the detection of an earthquake.

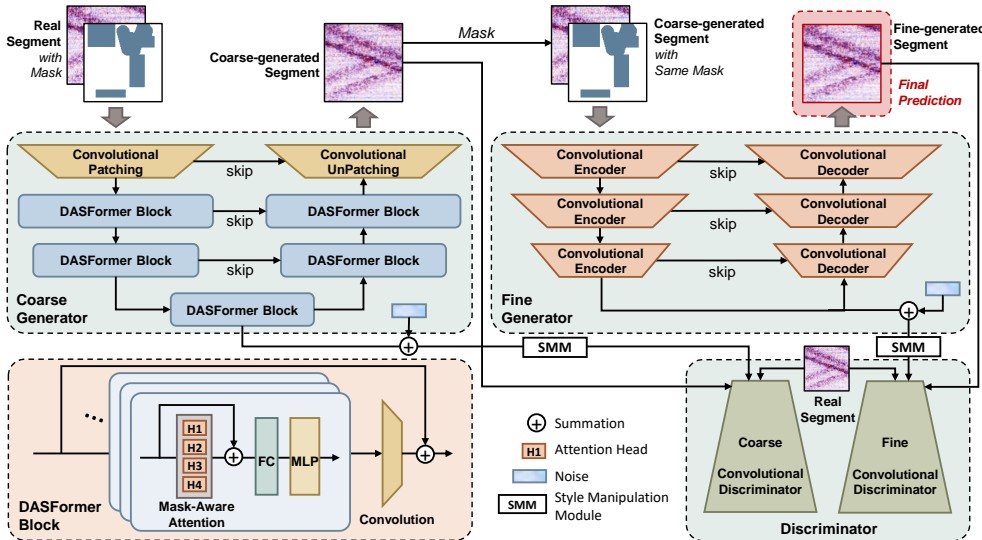

Figure 2: The pipeline of DASFormer with the coarse-to-fine framework including (i) Coarse Generator instantiated by Swin U-Net networks with DASFormer blocks, (ii) Fine Generator instantiated by Convolutional U-Net networks, and (iii) Noise-injected Discriminator instantiated by CNNs. For simplicity, here we omit patch-merging and patch-upsampling blocks between DASFormer blocks.

# 4 DASFORMER

In this section, we introduce a self-supervised pretraining scheme on DAS data named DASFormer. Our approach is basically built on the top of Swin U-Net Liu et al. (2021b); Cao et al. (2022) and Convolutional U-Net Ronneberger et al. (2015), and trained on the mask prediction task to learn the spatial-temporal representation of DAS data.

## 4.1 DAS DATA PREPROCESSING

DAS data can be typically represented as a time series with a large number of continuous variates (1250 in our datasets), which could be resource-consuming if we directly feed them into the model. To address this issue, we split the DAS data into segments based on its spatial-temporal invariance: (i) spatially uniform DAS sensor spacing and (ii) regular temporal sampling rate. Specifically, the raw DAS data is split into segments in size $V \times L$ with a stride of $\frac{V}{2}$ and $\frac{L}{2}$ alongside time and space dimensions, respectively, so that half of every two adjacent small blocks overlap.

## 4.2 ARCHITECTURAL DESIGN OF DASFORMER

The pipeline of our DASFormer model is illustrated in Figure 2.

**Part 1: Multi-variate Time Series Masking** Inspired by pretrained language models like BERT Devlin et al. (2019), we mask a certain percentage of the input DAS data across both the temporal and variate dimensions in multi-variate time series. The goal is to learn robust and generalizable time series representations by predicting the masked data points from the context of the remaining data. Practically, we obscure values using a combination of masks with both regular and irregular shapes (see the input in Figure 2). Please refer to Appendix C.1 for more details.

Specifically, given a segment $\mathbf{X}$ in shape $V \times L$, our masking strategy entails selecting a data subset and applying a mask. This mask, a binary matrix $\mathbf{M}$ with the same shape $V \times L$ as the input segment $\mathbf{X}$, assigns the value of 1 to masked data points and 0 to unmasked points. Thus, our input comprises the segment $\mathbf{X}$ and the corresponding mask $\mathbf{M}$.

**Part 2: Overlapping Convolutional Patching** The non-overlapping linear patching strategy in the standard Swin Transformer is unsuitable for multi-variate time series data due to its neglect of

temporal dependencies between adjacent patches, potentially disrupting local neighboring structures, particularly when discriminative regions are divided. To rectify this, we instead employ 2D convolutions to generate overlapping feature maps as patches, thereby capturing both temporal and spatial relationships. In this way, the input resolutions are not required to be strictly divisible by the predetermined patch size as the linear patching strategy. Formally, given the input in shape $V \times L$, we use two convolutional layers and produce $\frac{1}{2}$ sized feature maps as our patches.

**Part 3: Coarse-to-Fine Multi-variate Time Series Generator**   The DASFormer generator, designed to accurately predict missing data points, comprises two stages: a Swin U-Net-based network for coarse prediction and a U-Net network for fine prediction. This coarse-to-fine framework provides key advantages: (i) The coarse stage focuses on high-level features like contextual interaction in both temporal and spatial dimensions, while the fine stage refines predictions based on mask-specific detailed information, resulting in more realistic pretraining outcomes. (ii) The contextual knowledge acquired from the coarse stage can be transferred to downstream tasks, negating the need for training from scratch.

*DASFormer Block.* To enable the model to dynamically update the mask state based on the unmasked data points, we propose the DASFormer block to adapt DAS time series data. Specifically, we first modify the multi-head attention mechanism in the vanilla Swin-Transformer block to let tokens only attend to the unmasked tokens. Additionally, the global nature of the vanilla attention mechanism lacks the local temporal dependencies in time series Li et al. (2019b); Xu et al. (2022), so we propose an additional mask $\mathbf{B}$ with Gaussian distribution to enhance the locality perception of the attention block. Formally, the attention in our DASFormer can be formulated as follows:

$$\mathbf{a}_h = \text{softmax}\left[(\frac{\mathbf{Q}_h \mathbf{K}_h^\top}{\sqrt{d_k}}) + \widetilde{\mathbf{M}} + \mathbf{B}_h\right]\mathbf{V}_h, \quad \widetilde{\mathbf{M}}_{i,j} = \begin{cases} 0 & \text{if } \mathbf{M}_{\text{token}\,i,j} = 0 \\ -\text{inf} & \text{if } \mathbf{M}_{\text{token}\,i,j} = 1 \end{cases}, \quad (1)$$

where $h = 1, ..., H$ denotes the $h$-th head. $\mathbf{Q}_h$, $\mathbf{K}_h$, $\mathbf{V}_h$ denote Query, Key, and Value embeddings, respectively. $d_k$ denotes the dimension of embedding vectors. $\mathbf{M}_{\text{token}\,i,j}$ denotes the mask, and $\widetilde{\mathbf{M}}$ denotes the corresponding attention biases. $\mathbf{B}$ denotes the additional mask with Gaussian distribution $(\mathbf{B}_h)_{i,j} = -\frac{d(p(i),p(j))}{2\sigma_h^2}$, where $p(i) = (i_x, i_y)$ the 2-D absolute position of token $i$, and $d(p(i), p(j)) = (i_x - j_x)^2$ denotes 1-D Euclidean distance representing time distance between $i$ and $j$. The $\sigma_h$ for each head of attention can be different to learn multi-scale temporal dependencies. Please refer to Appendix C.3 for more details.

*Dynamic Masking.* Let $\mathbf{M}_{\text{token}}$ denote the token mask, and we define the initial token mask as $\mathbf{M}_{\text{token}}^{(0)} = \mathbf{M}$. In each training step, $\mathbf{M}_{\text{token}}$ is dynamically updated by an alternating strategy until all tokens are exposed as unmasked, meaning all masked tokens are learned and predicted. Specifically, in the $m$-th update, the mask $\mathbf{M}_{\text{token}}^{(m)}$ is obtained by applying a convolutional min-pooling over the recovered 2-D space of previous token mask $\mathbf{M}_{\text{token}}^{(m-1)}$ using a kernel in size $K \times K$ (with $K$ being even) with stride $K$. In the next update, an extra $K/2$ padding size is applied to have overlapping update pairs, which make the masked region gradually shrink and eventually disappear, i.e. eventually $\mathbf{M}_{\text{token}} = \mathbf{0}$. Please refer to Appendix C.2 for more details.

*Fine Generator.* We instantiate the fine generator with vanilla convolutional U-Net networks Ronneberger et al. (2015), which takes the predicted segment from the coarse generator as input.

**Part 4: Noise-injected Discriminator**   DAS data includes various stochastic noise, such as environmental and instrumental noise. To alleviate this stochastic variation and encourage more pluralistic generation, we inject stochastic noise into (i) the features after each convolution layer, and (ii) the weights of each convolution layer.

Specifically, the discriminator in DASFormer is basically instantiated by stacked convolution layers. Inspired by StyleGAN family Karras et al. (2019; 2020), we utilize a style manipulation module to reduce the sensitivity of the model to noisy inputs and encourage more diverse generations:

$$\boldsymbol{s} = f(\boldsymbol{C} + \boldsymbol{n}_c), \quad \boldsymbol{w}'_{ijk} = s_i \cdot \boldsymbol{w}_{ijk}, \quad \boldsymbol{w}''_{ijk} = \boldsymbol{w}'_{ijk} \left/ \sqrt{\sum_{i,k} {\boldsymbol{w}'_{ijk}}^2 + \epsilon} \right., \quad (2)$$

where we inject a noise $\boldsymbol{n}_c \sim \mathcal{N}(0, \sigma_c^2 \mathbf{I})$ to the code $\boldsymbol{C}$ learned from the generator.

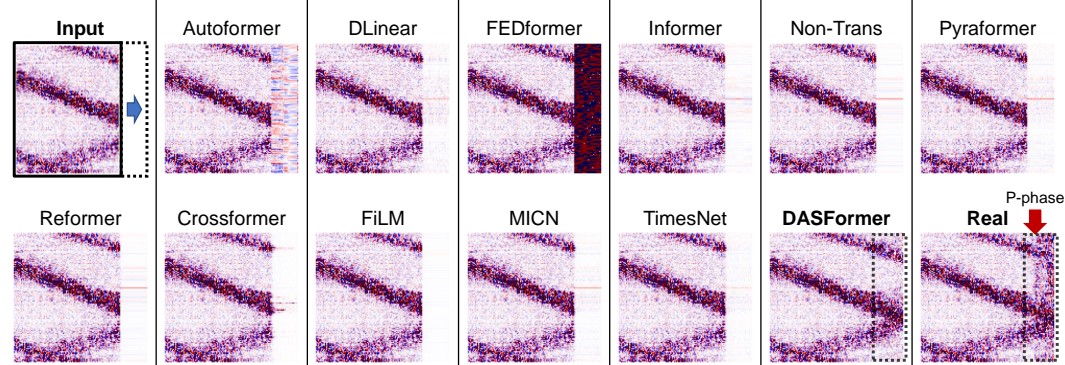

Figure 3: Visual comparison between forecasting methods. The dashed frame and the blue arrow denote the forecasting window and direction, respectively. The red arrow in the **Real** panel indicates an earthquake signal (P-phase) in real data. Note that our DASFormer successfully predicts the temporal trends of background signals while all baselines fail. Therefore, the earthquake can be easily exposed by measuring the distance between the forecasting region of **Real** and **DASFormer** panels.

**Objective Function**    Due to the presence of random noise in DAS data, it is not appropriate to evaluate the generation quality by measuring the point-wise error like MSE. Therefore, we aim to minimize the error of high-level feature maps instead of raw signals as the training objective of the generator. Specifically, we trained an extra VGG-based autoencoder Simonyan & Zisserman (2014) on our DAS data with reconstruction loss. Then we utilize its encoder as our high-level feature extractor for DAS data. Finally, the training objective of both coarse and fine generators is:

$$\underset{\theta_g}{\operatorname{argmin}} \mathcal{L}(\theta_g, \mathbf{X}) + \alpha||\nabla_{\mathbf{X}}\mathcal{L}(\theta_g, \mathbf{X})|| + \beta\mathcal{L}_G + ||\theta_g|| \tag{3}$$

$$\text{where } \mathcal{L}(\theta_g, \mathbf{X}) = \sum_i ||\phi_i(\hat{\mathbf{X}}) - \phi_i(\mathbf{X})||_1 \tag{4}$$

where $\phi_i(\cdot)$ denotes the $i$-th feature map extracted from the pretrained encoder, $\theta_g$ denotes the parameters of the generator, $\hat{\mathbf{X}}$ denotes the generated segment. $||\nabla_{\mathbf{X}}\mathcal{L}(\theta_g, \mathbf{X})||$ denotes the regularization trick on the gradient Mescheder et al. (2018), and $\mathcal{L}_G$ is the adversarial loss for generator in GAN Goodfellow et al. (2020). The loss for discriminator is the same as GAN.

### 4.3    APPLY DASFORMER TO DOWNSTREAM TASKS

**Earthquake Detection**    DAS signals comprise ubiquitous, uniform random noise and environmental noises like traffic, which exhibit explicit temporal patterns due to repetitive or cyclical processes. Conversely, P and S signals are influenced by unpredictable events like earthquakes, resulting in complex and irregular patterns. We can directly utilize DASFormer as a forecasting-based anomaly detector, capable of identifying irregular patterns that significantly deviate from predicted values, especially in the absence of ground-truth earthquake event labels.

To derive predicted values, we introduce a causal mask (only mask the tail $p$ time steps of the segment, see inputs in Figure 3). To have causal forecasting, we shift a 1-D window in size $K$ alongside only the time axis instead of the 2-D window in dynamic masking.

**Other Downstream Tasks**    As a pre-trained model, DASFormer can be naturally fine-tuned to adapt to arbitrary downstream tasks. Typically, we take the coarse part of DASFormer, the Swin U-Net-based blocks, as the feature extractor, and fine-tune the subsequent convolutions. Take the task of precise (point-level prediction) P/S phase picking as an example, we first freeze the blocks, then change the dimension of the output layers and apply the point-wise classification loss function, and finally re-train the DASFormer in a supervised-learning setting.

Table 1: Comparison results between state-of-the-art methods of time series forecasting, anomaly detection, and foundation models. We measure the distance by absolute error (AE), Earth-Mover Distance (EMD), and sliced EMD. For forecasting methods, the look-back and forecasting window sizes are set to 100 and 10, respectively. Method-specific denotes the distance proposed by the specific method. For reconstruction methods, we reconstruct all 110 time steps and use the last 10 for distance measuring. We use ROC-AUC score (AUC) and F1 score (F1) as metrics. The best and second-best results are in **bold** and underlined, respectively. '-' denotes not available. To save space, we select and present representative models here. For complete results, please refer to Table 6 in Appendix.

| Methods | Distance | AE | | EMD | | sliced EMD | | Method-specific | |
|---|---|---|---|---|---|---|---|---|---|
| | Metric | AUC | F1 | AUC | F1 | AUC | F1 | AUC | F1 |
| Traditional method | Aggregation-0 | 0.730 | 0.246 | 0.692 | 0.277 | 0.709 | 0.246 | - | - |
| | Aggregation-inf | 0.509 | 0.014 | 0.501 | 0.006 | 0.497 | 0.009 | - | - |
| Forecasting | LSTM | 0.726 | 0.243 | 0.691 | 0.277 | 0.709 | 0.253 | - | - |
| | TCN | 0.730 | 0.240 | 0.701 | 0.277 | 0.700 | 0.253 | - | - |
| | Transformer | 0.736 | 0.245 | 0.695 | 0.276 | 0.702 | 0.251 | - | - |
| | FEDformer | 0.522 | 0.071 | 0.492 | 0.130 | 0.508 | 0.131 | - | - |
| | DLinear | 0.729 | 0.233 | 0.690 | 0.272 | 0.703 | 0.247 | - | - |
| | Autoformer | 0.727 | 0.233 | 0.691 | 0.265 | 0.707 | 0.246 | - | - |
| Anomaly detection | LSTM-VAE | 0.727 | 0.241 | 0.698 | 0.273 | 0.707 | 0.251 | - | - |
| | ConvNet-GAN | 0.724 | 0.244 | 0.695 | 0.270 | 0.707 | 0.249 | - | - |
| | Deep-SVDD | 0.726 | 0.234 | 0.694 | 0.277 | 0.702 | 0.251 | 0.564 | 0.176 |
| | GDN | 0.732 | 0.247 | 0.698 | 0.279 | 0.708 | 0.255 | 0.618 | 0.114 |
| | Anomaly-Trans | 0.730 | 0.245 | 0.694 | 0.275 | 0.710 | 0.252 | 0.543 | 0.156 |
| Foundation model | TimesNet | 0.733 | 0.243 | 0.699 | 0.277 | 0.708 | 0.255 | - | - |
| | $j$-DAS | 0.754 | 0.249 | 0.703 | 0.272 | 0.698 | 0.263 | - | - |
| | Dasformer (Ours) | 0.886 | 0.552 | 0.890 | 0.527 | **0.906** | **0.565** | - | - |

## 5 EXPERIMENTS

### 5.1 EXPERIMENTAL SETUP

**Datasets** We conduct experiments using DAS data collected by Caltech in Ridgecrest city, California Li et al. (2021a) as shown in Figure 1, of which one month of data is publicly available via Southern California Earthquake Data Center Center (2013)[1]. The Ridgecrest DAS transforms a 10 km telecommunication cable into 1250 sensors in 8 m spacing and records the continuous data at 250 Hz sampling rate. We follow the filtering and cleaning procedure introduced in Zhu & Beroza (2019) to preprocess the raw signals. We split 90 earthquake events into training/validation/testing sets alongside the timeline with 45/20/25 events, respectively. Then we downsample the cleaned signals to 10Hz and cut the DAS signals into $128 \times 128$ segments with a stride of 64 steps using the preprocessing scheme aforementioned in Section 4.1. After that, we obtain 31,464 segments in total and 21,672/4,352/5,440 segments in training/validation/testing sets pertaining on the mask prediction task. The segmentation for evaluating the performance on the P/S phase detection task is different and will be introduced later.

**Benchmarks** To demonstrate the superiority of the proposed DASFormer method, we extensively select 12 and 7 recent state-of-the-art methods on time series forecasting and anomaly detection tasks, respectively, as the benchmarks in this paper, including LSTM Hochreiter & Schmidhuber (1997b), TCN Lea et al. (2017), Transformer Vaswani et al. (2017), Crossformer Zhang & Yan (2023), Reformer Kitaev et al. (2020), Pyraformer Liu et al. (2021a), Nonstationary Transformer (Non-Trans) Liu et al. (2022), Informer Zhou et al. (2021), FEDformer Zhou et al. (2022b), DLinear Zeng et al. (2022), Autoformer Wu et al. (2021), MICN Wang et al. (2023), and FiLM Zhou et al. (2022a) for forecasting-based methods, and LSTM-VAE Lin et al. (2020), LSTM-GAN Goodfellow et al. (2014), ConvNet-VAE Kingma & Welling (2014), ConvNet-GAN Goodfellow et al. (2014), Deep-SVDD Ruff et al. (2018b), GDN Deng & Hooi (2021), Anomaly Transformer (Anomaly-Trans) Xu et al. (2022)

---

[1]Ridgecrest DAS data is available on SCEDC AWS S3 bucket `s3://scedc-pds/Ridgecrest_DAS`

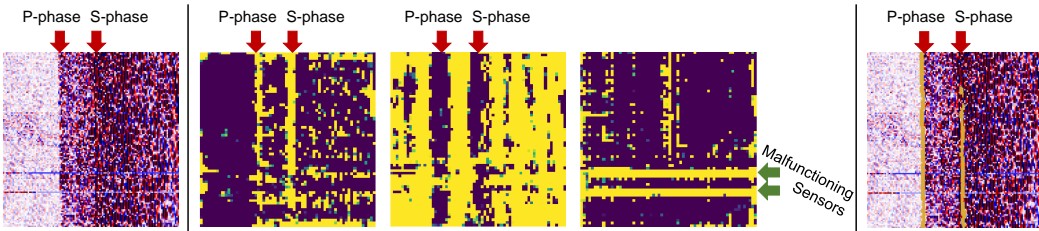

Figure 4: **(Left)** The input DAS signals. **(Middle)** Three selected feature maps extracted from the last DASFormer block of the pretrained DASFormer before fine-tuning. **(Right)** The P/S phase picking results of DASFormer in a semi-supervised setting trained with only 20 labeled samples.

for anomaly detection-based methods. For forecasting-based methods, we use the distance between the predicted values and the real values in the forecasting window as the anomaly score. For anomaly detection-based methods, the anomaly score is the distance between the reconstructed values and the observed values in the forecasting window. For Deep-SVDD, GDN, and Anomaly Transformer, we further show the results with their specific distances as anomaly scores. We involve two foundation models, TimesNet Wu et al. (2022) and $j$-DAS van den Ende et al. (2021b) and two traditional methods in Zhu & Beroza (2019).

**Implementation Details & Metrics** For earthquake detection task, we adopt a shift window with a smaller stride of 10 time steps (1 second) and predict the next 10 time steps to have non-overlapped forecasting alongside the time axis. We use window-wise labels instead of point-wise labels for a more robust evaluation. A forecasting region is designated as an anomaly if it contains at least one annotated anomaly data point. An anomaly score is defined by the distance between the realistic signal values and the predicted signal values within the forecasting region. We investigate several distance functions in practice, including absolute error with Euclidean distance (AE), Earth Mover's Distance (EMD) Rubner et al. (2000), and sliced EMD Deshpande et al. (2018). We calculate metrics, such as ROC-AUC and F1-score, for this binary classification.

For P/S phase picking task, we adopt convolutions on the top of the DASFormer blocks of the coarse generator and then fine-tune these convolutions with a point-wise loss function on the labeled DAS data. To make the model more robust to the signals, following Zhu & Beroza (2019), we soften the point-wise hard-encoded labels (1 for P/S phases, 0 for others) to values ranging from 0 to 1 by a Gaussian distribution.

## 5.2 RESULTS AND ANALYSIS

**Comparison Results** We compare our method with the aforementioned baselines on the earthquake detection task. Results in Table 1 suggest an overwhelmingly high ROC-AUC score and F1 score achieved by DASFormer. We also notice that AE and sliced EMD are both appropriate distance functions for this task. The visual comparison of forecasting-based methods is in Figure 3, where we can observe all baselines fail to forecast DAS signals. Reasons follow. (i) *Noisy*. DAS data can be quite noisy, with many spurious and random signals that don't correspond to any predictable events, making it difficult to train through point-wise loss used in all baselines. (ii) *Lack of spatial awareness*. All of these baselines treat different variates as independents, which ignores the spatial coherence of DAS data. Some of them utilize GNN to learn such correlations, however, the graph cannot accurately capture the way seismic signals propagate along the spatial axis, which is implied in the order of variates.

Reconstruction-based methods also failed for the following reasons. (i) *Lack of causal awareness*. Causal relations are crucial because we detect the anomalies based on the assumption that P/S phases are inherently unpredictable along the time axis. However, these methods ignore time direction and causality when reconstructing signals. (ii) *Scalable anomaly*. The generative methods such as ConvAE, VAE, and GAN are only sensitive to small abnormal areas. However, P-phase and S-phase can last for a long time, resulting in a large abnormal area.

Table 2: The results for P and S phases separately. The best results for each phase are in **bold**.

| Phase | AE | | EMD | | sliced EMD | |
|---|---|---|---|---|---|---|
| | AUC | F1 | AUC | F1 | AUC | F1 |
| P-phase | **0.860** | **0.335** | 0.830 | 0.302 | 0.843 | 0.327 |
| S-phase | 0.908 | 0.577 | 0.947 | 0.561 | **0.959** | **0.586** |
| Both | 0.886 | 0.552 | 0.890 | 0.527 | **0.906** | **0.565** |

Table 3: Comparison results for ablation studies.

| | AE | | EMD | | sliced EMD | |
|---|---|---|---|---|---|---|
| | AUC | F1 | AUC | F1 | AUC | F1 |
| Replace convolutional patching w/ linear patching | 0.826 | 0.435 | 0.770 | 0.341 | 0.663 | 0.299 |
| Replace DASFormer w/ Swin-Trans block | 0.869 | 0.428 | 0.782 | 0.335 | 0.748 | 0.316 |
| Replace high-level loss w/ Point-wise MSE loss | 0.880 | 0.435 | 0.839 | 0.499 | 0.743 | 0.314 |
| Remove noise injection in Discriminator | 0.856 | 0.523 | 0.857 | **0.540** | 0.830 | 0.443 |
| DASFormer (Ours) | **0.886** | **0.552** | **0.890** | 0.527 | **0.906** | **0.565** |

Benefiting from (i) the high-level reconstruction loss, (ii) the ability of Swin U-Net and convolutional U-Net to learn spatio-temporal patterns, and (iii) the causal forecasting-based anomaly detection mechanism, our DASFormer overcomes all of the above challenges.

**S/P Phase Analysis**   The results of DASFormer for P-phase and S-phase detection respectively are shown in Table 2. Our method performs better on S-phase than P-phase, which is within our expectations as earthquakes typically generate stronger S-phase due to their underlying physics. Intriguingly, AE and sliced EMD demonstrate superior performance on P-phase and S-phase, respectively. So in practice, we recommend the use of different distance functions for detecting P-phase and S-phase.

**More Downstream Tasks**   In Figure 4 (Left and Middle), we manually select and visualize 3 out of 180 feature maps from the outputs of the last DASFormer block in the pretrained coarse generator. These feature maps are surprisingly well-structured and informative, and already contain the patterns of P/S phases and malfunctioning sensors, demonstrating the fine-tuning ability of our DASFormer.

We attempt to fine-tune DASFormer for the precise P/S phase-picking task in a semi-supervised setting. We follow the data processing scheme in Zhu & Beroza (2019) which extends every picking to a region. Then we apply the simple linear probing fine-tuning scheme on the last two convolutional layers. A case study is shown in Figure 4 (Right). Moreover, the results of applying DASFormer to submarine DAS data are illustrated in Appendix D.3.

**Ablation Study**   To explore the role of each module in our proposed framework, we conduct ablation studies with the following comparisons: (i) convolutional patching v.s. linear patching, (ii) DASFormer block v.s. vanilla Swin Transformer blocks, and (iii) high-level encoder-based generative loss v.s. point-wise MSE generative loss. The results are shown in Table 3, where we conclude that all of the proposed modules could strengthen the performance of the model consistently.

# 6   CONCLUSION

This paper presents the DASFormer as a self-supervised framework for earthquake monitoring using DAS data. DASFormer leverages a well-designed coarse-to-fine framework with Swin U-Net and Convolutional U-Net architectures to effectively learn spatial and temporal correlations in DAS data. The proposed modules, such as convolutional patching, DASFormer blocks, and noise-injected discriminator, enable better adaptation of visual models to multi-variate time series data. Extensive comparisons demonstrate the superiority of our method in the task of unsupervised earthquake detection and other downstream tasks like semi-supervised P/S phase picking, submarine pattern extraction, etc. DASFormer opens the pathway for applying self-supervised learning to large-scale unlabeled seismic datasets for improving earthquake monitoring.

## 7    REPRODUCIBILITY STATEMENT

**Datasets**    We illustrate the detailed statistics of our datasets in Appendix B. The download link to one month of our data is given in the footnote in 5.1.

**Implementation Details**    We illustrate the implementation details in Appendix B, including benchmark description, hardware details, training setup, hyper-parameters setup, and training and inference speed.

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

## A  RESULTS WITH CONFIDENCE INTERVALS

Table 4 presents the average performances and $\pm 95\%$ confidence intervals for our results in Table 1 with five individual runs.

Table 4: Results with $\pm 95\%$ confidence intervals ($1.959 \times \sigma$).

| Distance | Metrics | |
|---|---|---|
| | AUC | F1 |
| AE | $0.886 \pm 0.0028$ | $0.552 \pm 0.0008$ |
| EMD | $0.890 \pm 0.0025$ | $0.527 \pm 0.0008$ |
| sliced EMD | $0.906 \pm 0.0039$ | $0.565 \pm 0.0006$ |

## B  IMPLEMENTATION DETAILS

**Datasets**  We only use segments in the testing set to evaluate the performance of DASFormer and all baselines. Unlike the pretraining, we adopt a shift window in $128 \times 128$ with a stride of 10 time steps (1 second) to obtain the segments for evaluation. After that, we have $14,688$ segments for evaluation.

We show the statistics of datasets in Table 5.

Table 5: Statistics of datasets.

| Descriptions | Training | Validation | Testing | Evaluation |
|---|---|---|---|---|
| # of seconds | 11370 | 1800 | 2250 | 2250 |
| # of sensors | 1250 | 1250 | 1250 | 1250 |
| Frequency rate | 10Hz | 10Hz | 10Hz | 10Hz |
| # of earthquakes | 45 | 20 | 25 | 25 |
| # of segments | 21672 | 4352 | 5440 | 14688 |
| Size of segments | $128 \times 128$ | $128 \times 128$ | $128 \times 128$ | $128 \times 128$ |
| Stride of segments | 64 | 64 | 64 | 10 |

Table 6: Complete comparison results between state-of-the-art methods of time series forecasting, anomaly detection, and foundation models. We measure the distance by absolute error (AE), Earth-Mover Distance (EMD), and sliced EMD. For forecasting methods, the look-back and forecasting window sizes are set to 100 and 10, respectively. Method-specific denotes the distance proposed by the specific method. For reconstruction methods, we reconstruct all 110 time steps and use the last 10 for distance measuring. We use ROC-AUC score (AUC) and F1 score (F1) as metrics. The best and second-best results are in **bold** and underlined, respectively. '-' denotes not available.

| Methods | Distance | AE | | EMD | | sliced EMD | | Method-specific | |
|---|---|---|---|---|---|---|---|---|---|
| | Metric | AUC | F1 | AUC | F1 | AUC | F1 | AUC | F1 |
| Traditional method | Aggregation-0 | 0.730 | 0.246 | 0.692 | 0.277 | 0.709 | 0.246 | - | - |
| | Aggregation-inf | 0.509 | 0.014 | 0.501 | 0.006 | 0.497 | 0.009 | - | - |
| Forecasting | LSTM | 0.726 | 0.243 | 0.691 | 0.277 | 0.709 | 0.253 | - | - |
| | TCN | 0.730 | 0.240 | 0.701 | 0.277 | 0.7 | 0.253 | - | - |
| | Transformer | 0.736 | 0.245 | 0.695 | 0.276 | 0.702 | 0.251 | - | - |
| | Crossformer | 0.730 | 0.246 | 0.702 | 0.275 | 0.712 | 0.251 | - | - |
| | Reformer | 0.730 | 0.247 | 0.694 | 0.277 | 0.709 | 0.253 | - | - |
| | Pyraformer | 0.726 | 0.245 | 0.693 | 0.278 | 0.707 | 0.254 | - | - |
| | Non-Trans | 0.727 | 0.238 | 0.697 | 0.277 | 0.710 | 0.252 | - | - |
| | Informer | 0.729 | 0.241 | 0.698 | 0.275 | 0.709 | 0.252 | - | - |
| | FEDformer | 0.522 | 0.071 | 0.492 | 0.130 | 0.508 | 0.131 | - | - |
| | DLinear | 0.729 | 0.233 | 0.690 | 0.272 | 0.703 | 0.247 | - | - |
| | Autoformer | 0.727 | 0.233 | 0.691 | 0.265 | 0.707 | 0.246 | - | - |
| | MICN | 0.739 | 0.247 | 0.699 | 0.277 | 0.705 | 0.256 | - | - |
| | FiLM | 0.735 | 0.245 | 0.694 | 0.278 | 0.702 | 0.250 | - | - |
| Anomaly detection | LSTM-VAE | 0.727 | 0.241 | 0.698 | 0.273 | 0.707 | 0.251 | - | - |
| | LSTM-GAN | 0.732 | 0.245 | 0.697 | 0.277 | 0.705 | 0.252 | - | - |
| | ConvNet-VAE | 0.734 | 0.247 | 0.701 | 0.280 | 0.711 | 0.257 | - | - |
| | ConvNet-GAN | 0.724 | 0.244 | 0.695 | 0.270 | 0.707 | 0.249 | - | - |
| | Deep-SVDD | 0.726 | 0.234 | 0.694 | 0.277 | 0.702 | 0.251 | 0.564 | 0.176 |
| | GDN | 0.732 | 0.247 | 0.698 | 0.279 | 0.708 | 0.255 | 0.618 | 0.114 |
| | Anomaly-Trans | 0.730 | 0.245 | 0.694 | 0.275 | 0.710 | 0.252 | 0.543 | 0.156 |
| Foundation model | TimesNet | 0.733 | 0.243 | 0.699 | 0.277 | 0.708 | 0.255 | - | - |
| | $j$-DAS | 0.754 | 0.249 | 0.703 | 0.272 | 0.698 | 0.263 | - | - |
| | Dasformer (Ours) | 0.886 | 0.552 | 0.890 | 0.527 | **0.906** | **0.565** | - | - |

**Benchmarks** We briefly introduce all baselines as follows:

- Aggregation-0 Zhu & Beroza (2019) averages the absolute values across all variables within a specific time window on DAS data.

- Aggregation-inf Zhu & Beroza (2019) serves like a random detector that assigns random labels without any specific pattern or criteria.

- LSTM Hochreiter & Schmidhuber (1997b), namely Long Short-Term Memory model, is a variant of recurrent neural networks (RNNs) that are capable of learning long-term dependencies.

- TCN Lea et al. (2017), namely Temporal Convolutional Networks, is a variant of convolutional neural networks (CNNs) that leverage convolutional operations to capture and model temporal dependencies in sequential data.

- Transformer Vaswani et al. (2017) embeds self-attention mechanism to capture cross-token dependency (cross-time dependency in time series) and demonstrates remarkable performance in sequence modeling tasks.

- Reformer Kitaev et al. (2020) is a variant of Transformer improving the computation efficiency by locality-sensitive hashing.

- Crossformer Zhang & Yan (2023) is a Transformer-based method utilizing cross-dimension dependency for multi-variate time series.

- Pyraformer Liu et al. (2021a) is a Transformer-based method learning multi-scale representation by the pyramidal attention module.

- Nonstationary Transformer Liu et al. (2022) is a Transformer-based method utilizing Series Stationarization and De-stationary Attention to address the over-stationarization issue for time series.

- Informer Zhou et al. (2021) is a Transformer-based method using ProbSparse self-attention to capture cross-time dependency for time series.

- FEDformer Zhou et al. (2022b) is a Transformer-based method utilizing the seasonal-trend decomposition to capture cross-time dependency for time series.

- DLinear Zeng et al. (2022) is a simple linear model with seasonal-trend decomposition for time series, which challenges Transformer-based methods.

- Autoformer Wu et al. (2021) is a Transformer-based method using the proposed Auto-correlation mechanism to capture cross-time dependency for time series.

- FiLM Zhou et al. (2022a), namely Frequency improved Legendre Memory model, proposes Legendre Polynomials projections to approximate historical information for time series.

- MICN Wang et al. (2023), namely Multi-scale Isometric Convolution Network, proposes isometric convolution for local-global correlations to capture the overall view (e.g., fluctuations, trends) for time series.

- LSTM-VAE Lin et al. (2020) and ConvNet-VAE Kingma & Welling (2014) are VAE-based generative methods using LSTM and convolutional networks as the encoder/decoder, respectively.

- LSTM-GAN Goodfellow et al. (2014) and ConvNet-GAN Goodfellow et al. (2014) are GAN-based generative methods using LSTM and convolutional networks as the generator, respectively.

- Deep-SVDD Ruff et al. (2018b) is an AE-based anomaly detection method built on a minimum volume estimation by finding a data-enclosing hypersphere to compute anomaly scores. We use convolutional networks as the backbone in experiments.

- GDN Deng & Hooi (2021) is an anomaly detection method utilizing graph neural networks to capture cross-dimension dependency for multi-variate time series.

- Anomaly Transformer Xu et al. (2022) is a Transformer-based anomaly detection method utilizing the proposed Anomaly-Attention mechanism to capture the cross-time association discrepancy for time series.

- $j$-DAS van den Ende et al. (2021b) is a foundation model with a self-supervised learning scheme for denoising of DAS data.

- TimesNet Wu et al. (2022) is a Transformer-based foundation model for time series, which achieves state-of-the-art performance in a variety of time series tasks.

The complete comparison results are shown in Table 6.

**Hardware**    All experiments are conducted on a Linux server with $4\times$ Nvidia RTX A5000 GPUs.

Table 7: The setup of hyper-parameters

| Notation | Value | Description |
| --- | --- | --- |
| $V$ | 128 | Number of variates in a segment. |
| $L$ | 128 | Number of time-steps in a segment. |
| $h$ | 8 | Number of heads in attention mechanism. |
| $K$ | 2 | Dimension of kernels for dynamic mask updating. |
| $\alpha$ | 1 | Trade-off hyperparameter in loss function. |
| $\beta$ | 0.1 | Trade-off hyperparameter in loss function. |

**Training Setup**    We use Adam Kingma & Ba (2015) optimizer with a learning rate of 0.001 and an early stopping strategy in the pre-training of our DASFormer model. The batch size of input segments is set to 32. We save the pre-trained model with the lowest loss on the validation set, which is evaluated on the testing set as an earthquake detector. The model is implemented using PyTorch Paszke et al. (2019). We show the setup of all hyper-parameters in Table 7. All hyperparameters are set empirically without careful tuning.

**Training & Inference Speed**    Typically, the pre-training of DASFormer takes around 72 hours with $4\times$ Nvidia RTX A5000 GPUs. The inference speed of causally forecasting DAS signals is around 0.02 seconds per segment with $1\times$ Nvidia RTX A5000 GPU.

## C    ILLUSTRATION OF MASKING

### C.1    REGULAR AND IRREGULAR MASKING

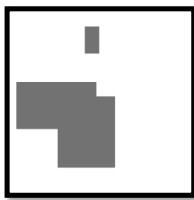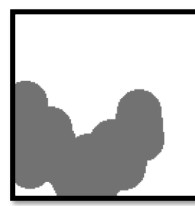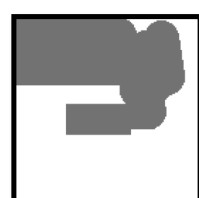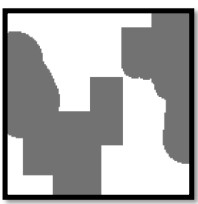

Figure 5: Samples of input masks used in DASFormer. From Left to Right: (i) regular-only mask, (ii) irregular-only mask, (iii) and (iv) a mix of regular and irregular masks. We use all these kinds of masks in practice.

We utilize the masking generation policy proposed by DeepFillv2 Yu et al. (2019) for segmentation in the pre-training stage. We visualize four samples of our regular and irregular masks in Figure 5.

### C.2    DYNAMIC MASKING

We illustrate an example of the alternating updating strategy for Dynamic Masking in Figure 6, where the entire updating can be done in two steps. In each step, we apply a convolutional min-pooling operation over the mask. Usually, the number of updates can be much more than two, if so, we repeat the first and second updates shown in Figure 6 in the following odd and even steps, respectively, until all tokens eventually become valid.

For forecasting, the causal updating strategy for Dynamic Masking is illustrated in Figure 7. By constantly applying 1D convolutional min-pooling operation over the mask, the model can eventually forecast all masked tokens.

### C.3    GAUSSIAN MASKING

We illustrate an example of Gaussian Masking in Figure 8, where the memory table stores the value of $\exp\left(-\frac{d(p(i),p(j))}{2\sigma_h^2}\right)$ at position $(p(i)_x, p(j)_x)$, where $p(i) = (i_x, i_y)$ the 2-D absolute position of

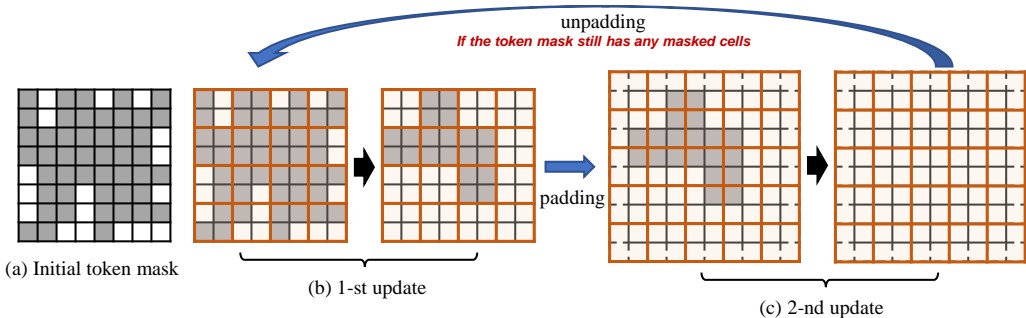

Figure 6: Illustration of the alternating updating strategy for Dynamic Masking in pre-training. The cells in grey and white mean masked and valid regions, respectively. For simplicity, the kernel size $K$ is set to 2 in this case. As we can see, all tokens are finally exposed as valid after 2 updates.

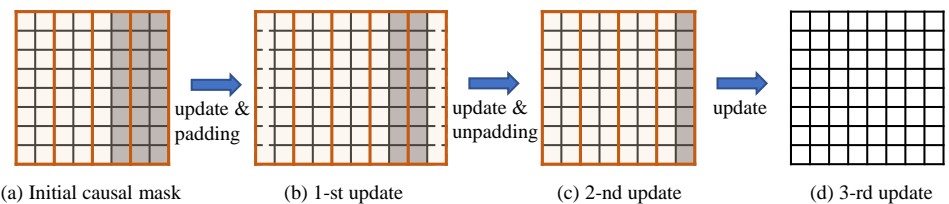

Figure 7: Illustration of the causal updating strategy for Dynamic Masking in forecasting. We shift a 1D window in size $K$ alongside the temporal axis to causally update the token mask.

token $i$, and $d(p(i), p(j)) = (i_x - j_x)^2$ denotes 1-D Euclidean distance representing time distance between $i$ and $j$. The value of Gaussian mask at position $(i, j)$ corresponds to the value of the memory table at position $(i_x, j_x)$. Empirically, we apply Gaussian Maskigng to 4 of 8 heads in our experiments, where the standard deviations $\sigma_h$ are set to $\{D/4, D/2, D, 2D\}$ for 4 heads, respectively, to learn multi-scale time dependencies, where $D$ is the size of the time axis of the recovered 2-D token space.

# D  CASE STUDIES

## D.1  ADDITIONAL CASES

We show eight additional cases in Figure 9. As we can see, DASFormer accurately captures the increasing or decreasing patterns of traffic flow over time, as well as artifacts caused by malfunctioning sensors. By successfully predicting both traffic trends and artifacts caused by broken sensors, DASFormer produces robust P/S phase detections for earthquake monitoring.

## D.2  FAILURE CASES

Recall that our labeling strategy is that a sample is labeled as an anomaly if at least one data point in the forecasting window is annotated as P/S phase. We show 2 failure cases in Figure 10. As we can see, the reasons for these failures are:

- Annotations are inaccurate.
- Annotations are ambiguous. Samples with insufficient earthquake signals in their forecasting windows are labeled as anomalies.

Therefore, some cases correctly detected by DASFormer are counted as false detections in the evaluation. This suggests that the practical performance of DASFormer should be better than what the metrics (i.e., ROC-AUC scores and F1 scores) reflect due to the inevitably inaccurate and ambiguous labeled samples.

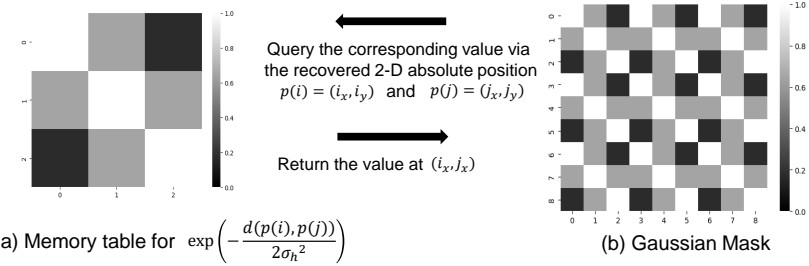

(a) Memory table for $\exp\left(-\dfrac{d(p(i), p(j))}{2\sigma_h^2}\right)$          (b) Gaussian Mask

Figure 8: Illustration of Gaussian mask for better locality perception. For simplicity, we take a token space in size $3 \times 3$ as an example, so the size of the Gaussian mask is $9 \times 9$ ($3^2 \times 3^2$). The standard deviation $\sigma_h$ in the Gaussian distribution is set to $3/2$.

### D.3    DASFORMER FOR SUBMARINE DAS DATA

Except for the Ridgecrest DAS example on land, DAS is also promising for deploying large-aperture and long-term monitoring networks at logistically challenging places. For instance, deploying DAS in the harsh ocean. We take advantage of a four-day community submarine DAS experiment offshore central Oregon Wilcock et al. (2023) to examine the robustness of our DASFormer. Between Nov. 1st and 5th, 2021, two fiber-optic backbone cables were temporarily converted to submarine DAS arrays (referred to as OOI North and OOI South) Wilcock et al. (2023). In this study, we use the OOI North as an example. It was connected to an Optasense QuantX interrogator to continuously record ground vibrations up to the first optical repeater located at ∼65 km from the shore. With a sensor spacing of ∼2 m, the OOI North array has a total of 32600 sensors[2].

We pre-train an Ocean-DASFormer on this submarine DAS data and visualize four cases in Figure 11:

- (Upper Left) Ocean gravity waves.
- (Upper Right) Land-ocean boundary (land on top and ocean at bottom).
- (Lower Left) and (Lower Right) Noise in the ocean at different water depths.

Ocean-DASFormer effectively captures the spatial and temporal patterns of all examples in various environments, indicating its potential for submarine DAS data in the future.

### D.4    TIME SERIES VIEW

We illustrate the results in a time series view in Figure 12. As we can see, DASFormer effectively captures the temporal trends and patterns for normal cases. And for anomaly cases, the difference between the forecast and real values becomes larger indicating an earthquake comes.

---

[2]The data is publicly available at http://piweb.ooirsn.uw.edu/das/

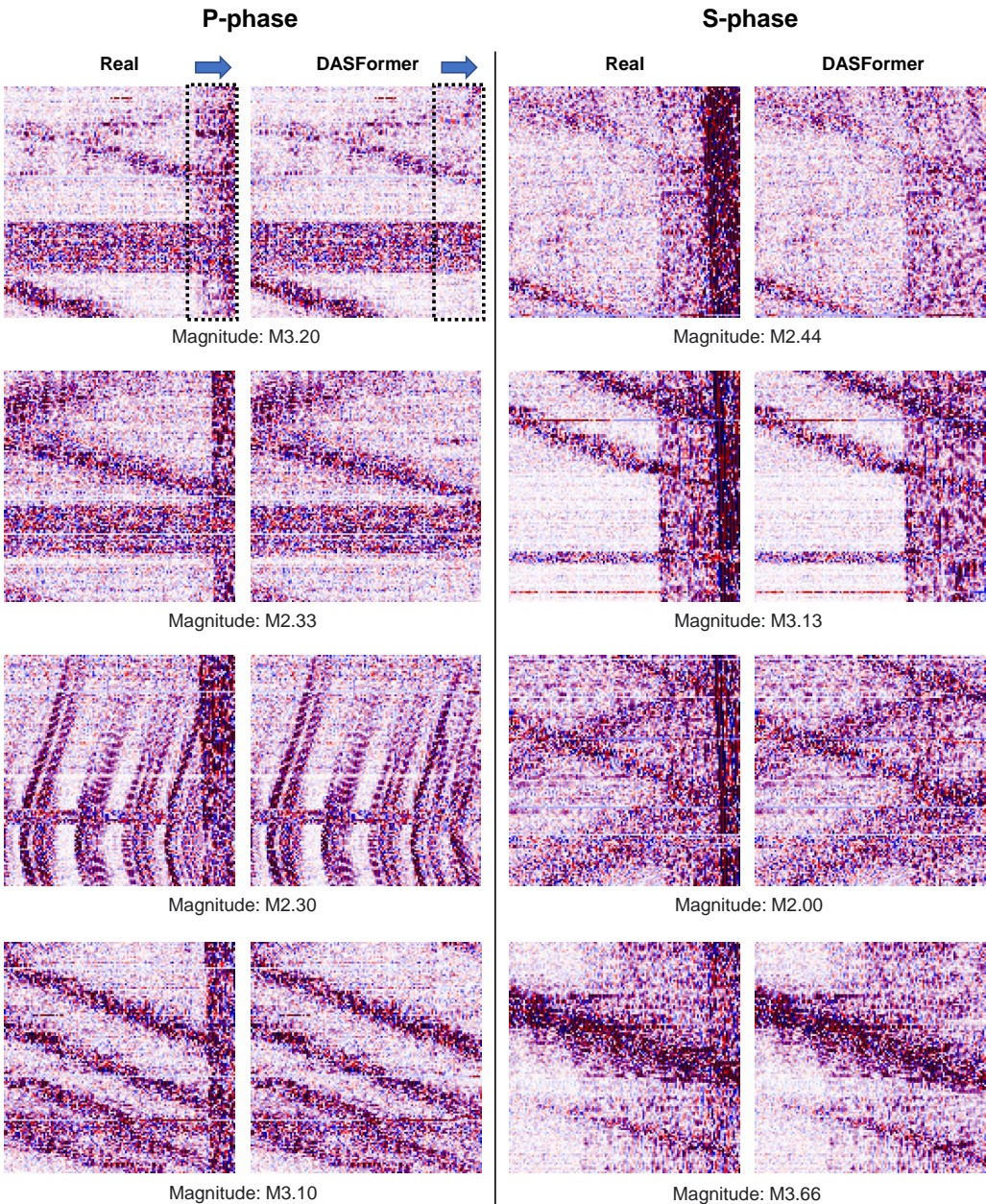

Figure 9: Visualization of forecasting results. The dashed frame and the blue arrow in the first case denote the forecasting window and direction, respectively, which are omitted in the rest of the cases for clarity. Our pre-trained DASFormer successfully detects all of these earthquake phases with high anomaly scores. Magnitude is from Southern California Earthquake Data Center https://scedc.caltech.edu/.

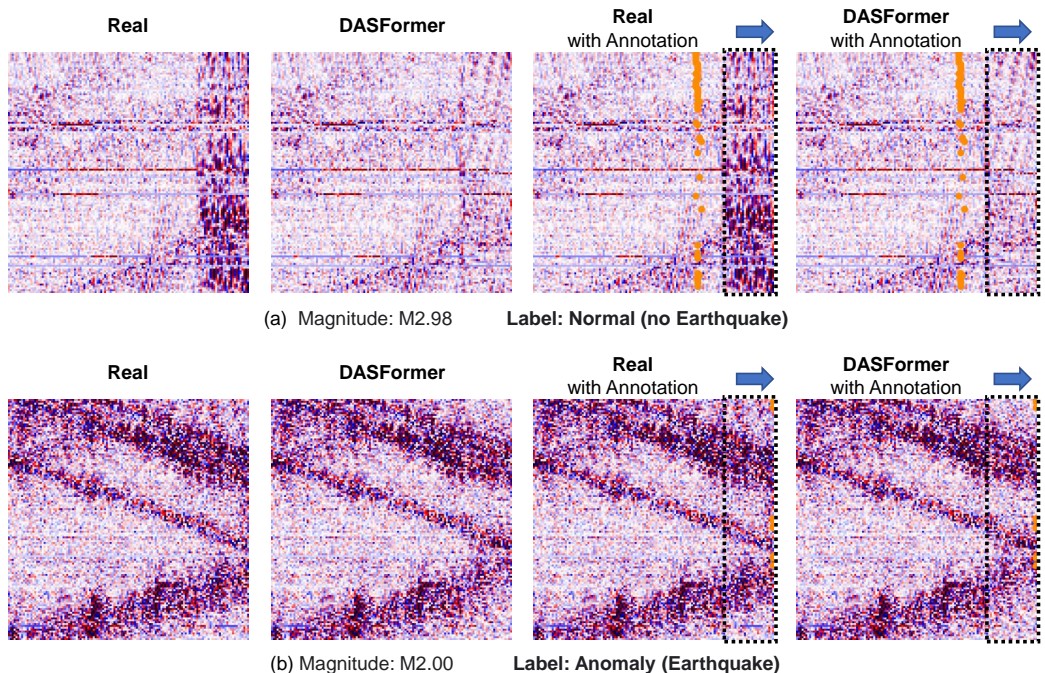

Figure 10: Visualization of failure cases. The dashed frame and the blue arrow in the first case denote the forecasting window and direction, respectively. The orange points indicate the data points annotated as P phase. (a) Inaccurate annotation. The sample should be labeled as an anomaly with earthquake. (b) Ambiguous annotation. The time period of earthquake signals in this sample is negligible to be labeled as an anomaly with earthquake.

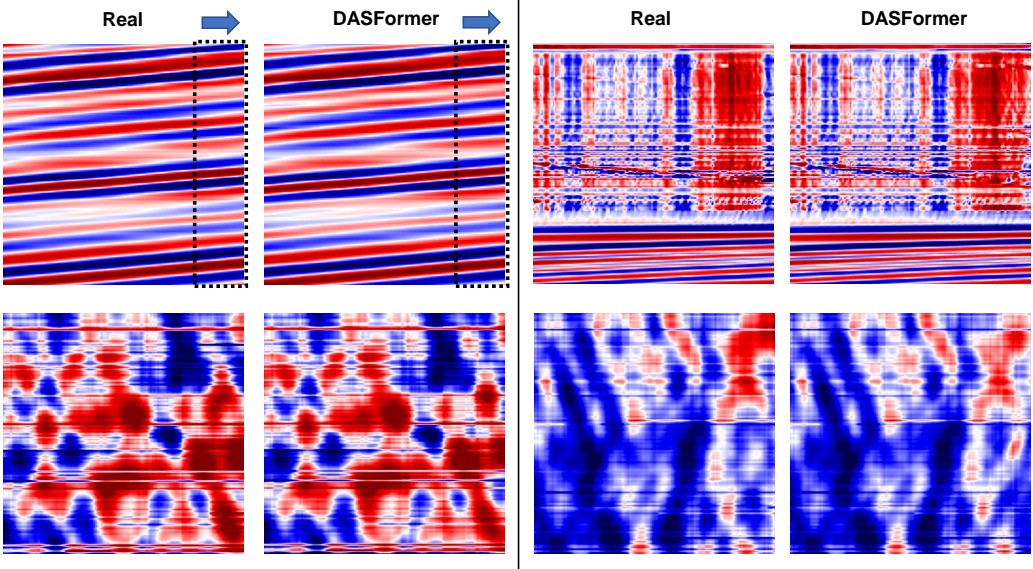

Figure 11: Visualization of applying DASFormer to submarine DAS data. The dashed frame and the blue arrow in the first case denote the forecasting window and direction, respectively, which are omitted in the rest of cases for clarity.

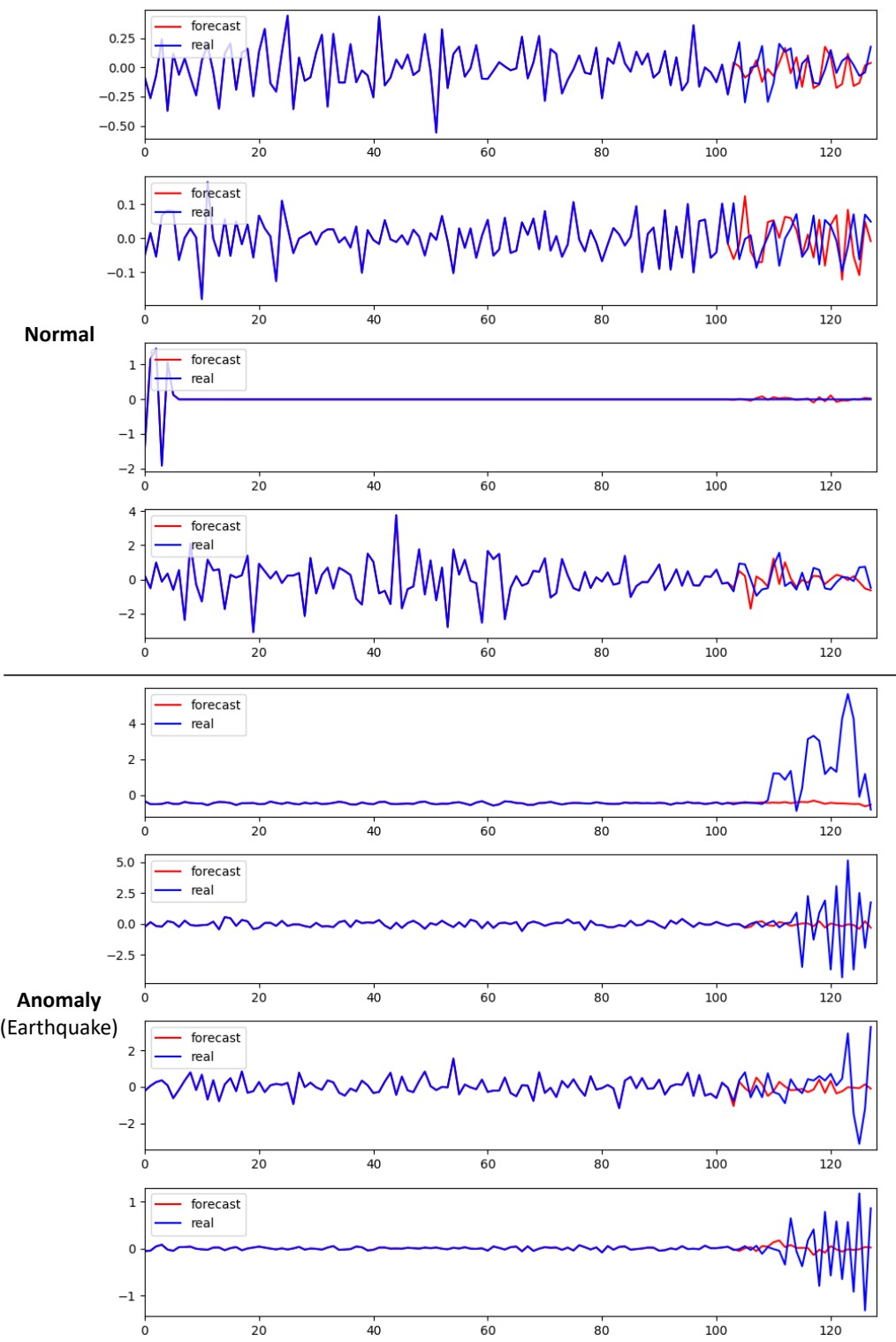

Figure 12: Illustration of results in a time series view. We randomly select 4 channels from normal cases and anomaly cases, respectively. The look-back and forecasting sizes are set to 103 and 25, respectively.

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
