# OpenReview forum: "DASFormer: Self-supervised Pretraining for Earthquake Monitoring"
_ICLR.cc/2024/Conference — Submitted to ICLR 2024_

### Official Review · Reviewer_gsBc · 2023-10-29

**Soundness:** 2 fair
**Presentation:** 3 good
**Contribution:** 3 good
**Rating:** 6
**Confidence:** 3

**Summary:**

The authors present a new self-supervised learning model for the task of earthquake monitoring given Distributed Acoustic Sensing (DAS) data. Specifically, the authors specify the evaluation as an anomaly detection task, where the DASFormer outperforms a wide range of various supervised models for detecting seismic phases. The details and motivation of the construction of the DASFormer, with the accompanying figure, are clear. The study adds to the existing literature of demonstrating the opportunities that DAS data has to offer.

**Strengths:**

The paper is clearly written and has a good structure that makes the paper nice to read. Enough details are given to understand the introduced DASFormer and the experimental setup. The evaluation on the seismic phase detection task includes an extensive range of supervised Deep Learning approaches, as well as a traditional baseline. On the downstream tasks the introduced SSL method shows significant improvement across metrics compared to the supervised methods.

**Weaknesses:**

Training Regime:

- The laborious objective function came a bit surprised and is the part that could benefit from an inclusion in the figure

Evaluation Scheme:

The main weakness I see in the paper, as I aim to layout below, is whether the selected methods DASFormer is compared to, are reasonable, or whether there could be other baselines that would perform stronger than the selected ones (even though they are regarded as SOTA)

- I understand your main claim of the paper to be that your choice of architecture and training scheme can achieve superior results to existing Deep learning Models. However, your evaluation scheme is mainly demonstrating the usefulness of SSL pretraining using a lot more data than the other supervised methods have available. Thus, to me the more relevant question is why is your extensive pretraining technique necessary as opposed to "more standard" SSL methods in Computer Vision like MAE, Moco etc. In other words you should compare your SSL technique to another SSL technique to support the claim that it is necessary and superior. Therefore, I would be highly interested in a "baseline" SSL technique that you can compare the DASFormer results to. I would believe that this could potentially also give more insights into the unique aspects of DAS data. Additionally, you could evaluate the quality of extracted features from different SSL methods via Linear Probing
- ignoring the DASFormer approach for a second, your results table shows that the "traditional" aggregation baseline "aggregation-0" is actually really strong, on par with or better than all the Deep Learning models, doesn't this illustrate that there could be potentially some other, much simpler modelling approach to achieve good results?
- There is some literature, for example [Makridakis et al 2020](https://www.tandfonline.com/doi/full/10.1080/01605682.2022.2118629) and [Nixtla](https://github.com/Nixtla/statsforecast/tree/main/experiments/m3), showing that simple models outperform Deep Learning methods on time-series tasks, so I wonder what results you would get with a Random Forest for example, that could also offer more insights into "explaining" a model prediction
- The authors point out several times that it is not appropriate to train models with the standard point-wise loss. However, this point-wise loss is applied in all the baseline models that the DASFormer is compared to, so shouldn't there be a comparison to a baseline that does not have this weakness?

General Comment:

- Given the crucial task of earthquake monitoring and the author's acknowledgement that "DAS data includes various stochastic noise, such as environmental and instrumental noise", I was very surprised to not find the word "uncertainty" or a discussion thereof anywhere in the paper. Uncertainty is of course not a trivial topic, especially with Deep Learning models that are so over parameterized, nevertheless there should be an effort to at least discuss its influences and impacts.

**Questions:**

Content:
- Under 5.1"Implementation Details" you state that the anomaly score is defined between the predicted model values and the actual values. How does your approach work in a "real" setting then when you are forecasting into regions without any data? Or is it enough due to the time-scale of seconds to detect an anomaly "in hindsight"? This was not entirely clear to me.
- Are there advantages/disadvantages of the selected anomaly score distance metrics that you could discuss?

Writing:
- First sentence of section 3, I think you mean "also known as" instead of "as known as"

---

> ### Author Response · Authors · 2023-11-23
>
> **Q1:** The evaluation scheme mainly demonstrates the usefulness of SSL pretraining with a large amount of data, but it is unclear why the extensive pretraining technique is necessary compared to more standard SSL methods in CV.
>
> **A1:** Thanks for your valuable suggestion. We have indeed considered representing DAS data as images, but drawing an analogy between DAS data and images can also be misleading. Image data does not inherently consider the causal information along the time axis that is present in DAS data. Adapting image-based methods to a causal-related forecasting-based approach would require additional design considerations. We will explore the possibilities like adapting ViT-based image methods in the future.
>
> ---
>
> **Q2:** The comparison table shows that the "traditional" aggregation baseline performs on par with or better than the deep learning models, suggesting the potential for simpler modeling approaches. There is literature showing that simple models can outperform deep learning methods on time-series tasks like Random Forest.
>
> **A2:** Thanks for pointing out this. Aggregation-0 in Table 1 is a variant of well-known traditional method "STA/LTA" for DAS data.
>
> The traditional time-series methods like Random Forest and ARIMA-family are not applicable to our task, as they are designed for stationary time series. However, DAS data is characterized by dynamic and evolving patterns, making it challenging to model using traditional stationary time series methods
>
> ---
>
> **Q3:** The paper does not include a comparison to a baseline that does not use the point-wise loss, despite mentioning its weakness in the baseline models.
>
> **A3:** Thanks for pointing out this. To alleviate the drawback of point-wise loss, DASFormer utilizes an extra pretrained VGG-based on intermediate features (it's a features-to-features autoencoder, not on the final predictions) in DASFormer to extract the high-level features, and use the feature-wise loss as the metric. This feature extractor is not universal for other methods, so we have to design feature extractors/encoders to suit the specific architecture and characteristics of each baseline model. Therefore, we only use the point-wise loss for other baselines.
>
> ---
>
> **Q4:** The paper lacks a discussion of uncertainty, which is important for earthquake monitoring given the stochastic noise in DAS data.
>
> **A4:** Thanks for your suggestion. We show the confidence intervals of the results in Table 4 in Appendix, where we can observe that the results are stable and reliable with small variance. Due to the limitations of time and computation, we will explore and add more uncertainty analysis in the future work.
>
> We also conduct comparison experiments for different magnitude (different noise levels) of earthquakes. The results are shown in the following tables:
>
> Distribution of the magnitude of the 45 earthquakes in the training set:
>
> | Magnitude | Numbers |
> |-----------|---------|
> | M0 ~ M1   | 0       |
> | M1 ~ M2   | 0       |
> | M2 ~ M3   | 31      |
> | M3 ~ M4   | 12      |
> | M4 ~ M5   | 2       |
>
> Distribution of the magnitude of the 45 earthquakes in the evaluation set and the performance:
>
> |DASFormer          |         | AE   |      | EMD  |      | sliced EMD |      |
> |-----------|---------|------|------|------|------|------------|------|
> | Magnitude | Numbers | AUC  | F1   | AUC  | F1   | AUC        | F1   |
> | M0 ~ M1   | 0       | N/A  | N/A  | N/A  | N/A  | N/A        | N/A  |
> | M1 ~ M2   | 1       | N/A  | N/A  | N/A  | N/A  | N/A        | N/A  |
> | M2 ~ M3   | 27      | 0.84 | 0.4  | 0.88 | 0.46 | 0.86       | 0.42 |
> | M3 ~ M4   | 14      | 0.89 | 0.59 | 0.92 | 0.58 | 0.93       | 0.6  |
> | M4 ~ M5   | 3       | N/A  | N/A  | N/A  | N/A  | N/A        | N/A  |
>
> |Aggregation-0 |         | AE   |      | EMD  |      | sliced EMD |      |
> |-----------|---------|------|------|------|------|------------|------|
> | Magnitude | Numbers | AUC  | F1   | AUC  | F1   | AUC        | F1   |
> | M0 ~ M1   | 0       | N/A  | N/A  | N/A  | N/A  | N/A        | N/A  |
> | M1 ~ M2   | 1       | N/A  | N/A  | N/A  | N/A  | N/A        | N/A  |
> | M2 ~ M3   | 27      | 0.74 | 0.24 | 0.7  | 0.28 | 0.7        | 0.25 |
> | M3 ~ M4   | 14      | 0.73 | 0.24 | 0.69 | 0.26 | 0.7        | 0.25 |
> | M4 ~ M5   | 3       | N/A  | N/A  | N/A  | N/A  | N/A        | N/A  |
>
>
> For the earthquakes in the range of M2 ~ M3 and M3 ~ M4, we can see that the performance is becoming better for larger magnitude earthquakes. In each level, the performance of DASFormer is better than Aggregation-0. For the earthquakes in the range of M0 ~ M2 and M4 ~ M5, the number of samples is too small to draw a conclusion.

---

> > ### Author Response · Authors · 2023-11-23
> >
> > **Q5:** How does the approach work in a real setting? Is it sufficient to detect anomalies in hindsight due to the time-scale of seconds?
> >
> > **A5:** In a real setting, our approach works by predicting the DAS sginals for a future time window, given the current time window. Specifically, at time $t$, our model predicts the DAS waveforms for the subsequent time window $t+ /delta t$. Once the true DAS waveforms for the $/delta t$ time period are received, we calculate the difference between the predicted waveforms and the actual waveforms as an anomaly score. Typically, the time window $/delta t$ is set to be around 1 second, which is sufficient for detecting anomalies in real-time.
> >
> > ---
> >
> > **Q6:** Are there advantages/disadvantages of the selected anomaly score distance metrics that could be discussed?
> >
> > **A6:** The main advantage of Earth Moving Distance (EMD) over Absolute Error (AE) is that EMD is more robust to temporal and spatial dynamics. EMD considers the shift or displacement of the predicted earthquake waveforms compared to the actual waveforms, while AE is a point-to-point error. However, EMD with a large window size can be more sensitive to noise and small local variations in the waveforms. To trade-off between EMD and AE, we propose to use sliced EMD, which breaks down the waveforms into smaller segments and aggregates EMDs of all segments, enabling a more localized evaluation.

---

### Official Review · Reviewer_Hht1 · 2023-10-30

**Soundness:** 3 good
**Presentation:** 3 good
**Contribution:** 2 fair
**Rating:** 5
**Confidence:** 4

**Summary:**

This paper presents a foundation model for DAS data (optic fiber), used in particular for earthquake detection. The specificities of the data (noise, time dependence, spatial dependence) are well studied in order to propose a specific fondation model; based on Swin-Unet, convolutional U-net and GAN-strategy. For the self-supervision, a masking strategy is applied. Comparisons with forecasting multi-variate time series and anomaly detection methods are performed.

**Strengths:**

The paper is correctly written, the method is interesting (even though very complex) and the forecasting results are convincing. But I have several major concerns that needs to be clarifed first.

**Weaknesses:**

1) The first concern is the goal of the paper. Indeed, DAS earthquake detectors exists (one of them was cited by the autors, PhaseNet-Das, Zhu et al. 2023, there might be others), and no comparison was made, nor a justification on the benefit of your method against theirs. If the claim is to say that this is a foundation model, and the test on this task is only as a proof of concept, it should be clearer, and then show or justify a future useful application.
2) I think the purpose of a foundation model would be its applicability at a larger scale. Yet, is your method generalizable to other DAS sensors? It is not clear whether it is site and sensor-specific or not; if so it means a new self-training needs to be performed again for any new DAS.
3) The whole idea of this method is that earthquakes are unpredictible. It is clever indeed, but I see 2 major limitations: 1) this foundation model is thus harder to use for other tasks (which could be predictable) 2) in a series of aftershocks (which could maybe be seen as more predictable), how does your measure performs?
4) The comparison with other multi-variate time series are somehow misleading. Indeed, in multi-variate time-series, we suppose that the different time series (or sensors) are not ordered and not equally-spaced: DAS is a very particular type of 'multi-variate time-series'. I don't think it is worth presenting all of these methods (maybe only one), and it should be clearly stated in the paper. Yet, a comparison with image 2D foundation models, or by modifying a video framework from a 2D+t to a 1D+t, would be more relevant.

**Questions:**

Other questions:

Q1) Masking: the authors said that it is inspired by BERT, but maybe closer to this application, by self-supervised on image data? like Pathak, Deepak, et al. "Context encoders: Feature learning by inpainting." CVPR 2016.(and later works with ViTs)
Q2) I don't understand the dynamic masking, and I don't see why it is useful. Does it comes from other studies?
Q3) The ablation study should first be 'ablation': i.e. what if we only use the fine generator, or the fine coarse? What if we only use a 'generator' without the need of the discriminator? Also, why is the fine generator a convolutional U-net, and the coarse a Swin-based Unet?
Q4) comparison with PhaseNet is rather unclear. First, it is stated that the table 1 shows comparisons with time series forecasting and anomaly detection, but PhaseNet is neither of those, and is only single-station. In the table, it is unclear the use of 'traditional method', and the 'aggregation'. Would have been best to compare with PhaseNet-DAS, no?

- Figure 1 left: sensors and time axes not clear.
- page 2 : define P and S
- Figure 2: what is smm?

---

> ### Author Response · Authors · 2023-11-23
>
> **Q1:** Lacks a comparison with other DAS earthquake detectors like PhaseNet-DAS.
>
> **A1:** To the best of our knowledge, the existing DAS earthquake detectors like PhaseNet-DAS are for P/S phase pickup tasks, which are different from seismic detection tasks we did. Please refer to the Answer to A6.
>
> P/S phase pickup refers to the identification and classification of seismic wave phases, specifically the arrival times of the P (primary) and S (secondary) waves, which focuses on accurately detecting and labeling these specific phases point-wise in DAS data.  On the other hand, seismic detection task (in Table 1) involves identifying and classifying anomalies or abnormal patterns in DAS data. The main differences between P/S phase pickup and seismic detection tasks can be summarized in the following aspects:
>
> (1) **Wave identification**: In P/S phase pickup, the method needs to accurately distinguish between P and S waves in seismic data. But seismic detection tasks do not require this specific wave identification, only with earthquake events. Most of the existing P/S phase pickup methods like PhaseNet-DAS learn on the context of P/S phases, while our method is a forecasting-based method which is easier and more suitable for real-time systems.
>
> (2) **Data representation**: P/S phase pickup typically provides point-wise results, indicating the exact arrival times of P and S waves. In contrast, seismic detection tasks focus on determining whether there is an event or anomaly within a specific time span, without the need for precise point-wise results. Moreover, our training/valid/testing splits are different from PhaseNet/PhaseNet-DAS.
>
> (3) **Learning setting**: P/S phase pickup methods are typically implemented in a supervised learning setting, where labeled data with ground-truth P/S wave information is required for training. However, our seismic detection task is addressed in an unsupervised learning setting like anomaly detection, as obtaining labeled data for anomalies can be challenging.
>
> As a self-supervised learning model, DASFomer can be adapted to the P/S phase pickup tasks and we've done some preliminary experiments shown in the paper. However, considering the above points and the limited ground-truth annotations of P/S phases of our data, we hardly find a fair evaluation metric to compare with PhaseNet-DAS. We will explore more in the future work.
>
> ---
>
> **Q2:** The generalizability of the method to other DAS sensors is not addressed, and it is unclear if the self-training needs to be performed again for different DAS setups.
>
> **A2:** Our method is applicable to any other DAS site. In the preprocessing stage, we downsample and resize the DAS data to match the resolution required by the model. By downsampling the data to the same resolution, our method can be easily transferred and applied to new DAS sensors. This approach is similar to computer vision models trained on ImageNet, where they are designed to be applicable to various image resolutions.
>
> ---
>
> **Q3:** The method's suitability for other tasks and its performance in predictable scenarios, such as aftershocks, is not discussed.
>
> **A3:**  The effectiveness of our model for unpredictable earthquakes is derived from its good capabilities for other predictable events.
>
> In this work, we primarily showcase the usage of DASFormer as a forecasting-based earthquake monitor. For tasks that are predictable, we can remove the masking and directly employ DASFormer as a feature extractor. We demonstrate its capability as a feature extractor in detecting malfunctioning sensors in Figure 4. Additionally, Figure 3 also illustrates its ability to detect predictable events such as traffic signals.
>
> Regarding your concern about the assumption that "earthquakes are unpredictable" may potentially confuse the model, it is important to note that our look-back window is only around 13 seconds, with a prediction window of approximately 1 second. At this time scale, the level of predictability is nearly indistinguishable for relatively predictable events like aftershocks. Thus, we can treat each aftershock as a separate earthquake event, considering them as individual instances rather than relying on their predictability.

---

> > ### Author Response · Authors · 2023-11-23
> >
> > **Q4:** The comparison with other multi-variate time series methods is misleading, as DAS data has unique characteristics that require a different approach. Comparisons with image-based models or modifications of video frameworks would be more relevant.
> >
> > **A4:** Thanks for your valuable suggestion. Actually we did conduct baseline experiments using image-related methods, such as CNN-GAN or CNN-VAE, shown in Table 1. We have indeed considered representing DAS data as images, but drawing an analogy between DAS data and images can also be misleading. Image data does not inherently consider the causal information along the time axis that is present in DAS data. Adapting image-based methods to a causal-related forecasting-based approach would require additional design considerations. We will explore the possibilities like adapting ViT-based image methods in the future.
> >
> > ---
> >
> > **Q5:** Masking: the authors said that it is inspired by BERT, but maybe closer to this application, by self-supervised on image data?
> >
> > **A5:** Yes, if we consider DAS data as images, we can adapt image-based self-supervised learning (SSL) methods for DAS data. In fact, our method is a combination of Swin Transformer and U-Net, which are two image-based methods, adapted specifically for DAS data. The masking technique used in our approach, inspired by BERT, is a way to incorporate self-supervised learning on the DAS data.
> >
> > ---
> >
> > **Q6:** What is the purpose and inspiration behind the dynamic masking used in the method?
> >
> > **A6:** The purpose of dynamic masking in our method serves two main functions.
> >
> > Firstly, dynamic masking ensures that only unmasked tokens are attended to within the self-attention blocks, while disregarding the masked (invalid, useless) tokens. In this way, the model can more effectively capture the temporal/spatial dependencies.
> >
> > Secondly, the dynamic masking approach updates the masked tokens around the unmasked tokens in a step-by-step manner, similar to a time series auto-regression rather than a one-step regression. This helps the model to progressively refine its predictions and incorporate the newly revealed information from the previously masked tokens.
> >
> > ---
> >
> > **Q7:** Could the ablation study explore different combinations of using only the fine generator, coarse generator, or a generator without the discriminator? Why are different network architectures used for the fine and coarse generators?
> >
> > **A7:** Yes, we can have coarse-only, coarse-and-fine, coarse-and-fine w/ discriminator, these three components. The results are shown in the following table:
> >
> > |                       | AE   |      | EMD  |      | sliced EMD |      |
> > |-----------------------|------|------|------|------|------------|------|
> > |                       | AUC  | F1   | AUC  | F1   | AUC        | F1   |
> > | Coarse-only           | 0.79 | 0.41 | 0.81 | 0.44 | 0.83       | 0.47 |
> > | Coarse-to-Fine        | 0.85 | 0.52 | 0.87 | 0.53 | 0.87       | 0.50 |
> > | Coarse-to-Fine w/ GAN | 0.88 | 0.55 | 0.89 | 0.52 | 0.90       | 0.56 |
> >
> > The reason to use different networks for two stages is that Swin Transformer is patch-level model, which can more effciently update the masked regions (see dynamic masking), while U-Net is a point-level model and more fine-grained.
> >
> > ---
> >
> >
> > **Q8:** The comparison with PhaseNet is unclear as it is not a time series forecasting or anomaly detection method. Comparing with PhaseNet-DAS would be more appropriate.
> >
> > **A8:** Please refer to A1.

---

### Official Review · Reviewer_RSrX · 2023-10-31

**Soundness:** 3 good
**Presentation:** 3 good
**Contribution:** 2 fair
**Rating:** 5
**Confidence:** 3

**Summary:**

Authors apply self-supervised learning to Distributed acoustic sensing (DAS) data to learn representations that are suitable for several downstream tasks.

**Strengths:**

**Originality.** While the application to DAS data might be original, the paper is essentially an application of self-supervised learning to a new domain.

**Quality and clarity.** The paper is well-written and easy to follow.

**Significance.** Results seem to be promising, lacks comprehensive sensitivity analysis to be a viable real-world solution.

**Weaknesses:**

* While the approach seems to outperform several baselines, the visual inspection of the forecasting results in the appendix is underwhelming.

* Detection becomes more important and challenging for smaller earthquakes. It is unclear how the method performs for smaller earthquakes and whether there are many false positives.

* I found this paper to be too focused on the application and not enough on the method. I feel there are not enough methodological novelties to make it a suitable choice for this conference.

**Questions:**

* It would be beneficial to conduct a sensitivity analysis of the method concerning the choice of hyperparameters, especially since this is a real-world application. As authors have indicated, this application is associated with hazards to public safety so quantifying the uncertainty of the method is crucial.

* What is the distribution (histogram) of the metric as a function of earthquake magnitude? How does it compare to the baseline? How about traditional signal processing techniques?

---

> ### Author Response · Authors · 2023-11-23
>
> **Q1:**  It is unclear how the method performs for smaller earthquakes and whether there are many false positives.
>
> **A1:** For performance on smaller earthquakes, we visualized several cases ranging from M2.0 to M3.66 in Figure 9 in Appendix. And we show the performance for different magnitudes in answer **A3**. The false positive rate (FPR) and true positive rate (TPR) for DASFormer are shown as follows:
>
> |     | AE   | EMD  | sliced EMD |
> |-----|------|------|------------|
> | FPR | 0.22 | 0.24 | 0.19       |
> | TPR | 0.79 | 0.82 | 0.81       |
>
> ---
>
> **Q2:** It would be beneficial to conduct a sensitivity analysis of the method concerning the choice of hyperparameters since this application is associated with hazards to public safety
>
> **A2:** Thanks for your suggestion. We have the results with confidence intervals in Table 4 in Appendix. Actually we didn't careflly tune these hyperparameters due to the time and computation limitation. The training time of DASFormer is about 3~4 days on 4 NVIDIA RTX A5000 GPUs. We list all the 7 hyperparameters and the empirical setup in Table 7 in Appendix, where we think only $\alpha$ and $\beta$ are sensitive to the performance. For the resolution $V$ and $L$, they are larger is better, and are set to $128$ which is the maximum under our computation. For the number of heads $h$, it is empirically set as in multi-head attention mechanism. For the dimension of kernels $K$, it is less is better, and $2$ is the minimum value that we can use. For $\alpha$ and $\beta$, they are the trade-off between the reconstruction loss and the normalization tricks, which are empirically set to $1$ and $0.1$.  We will conduct a sensitivity analysis of the method concerning the choice of hyperparameters once we have the results in the future. Thanks!
>
> ---
>
> **Q3:** What is the distribution (histogram) of the metric as a function of earthquake magnitude? How does it compare to the baseline? How about traditional signal processing techniques?
>
> **A3:** We provide the magnitude distribution of our datasets with performance here:
>
> Distribution of the magnitude of the 45 earthquakes in the training set:
>
> | Magnitude | Numbers |
> |-----------|---------|
> | M0 ~ M1   | 0       |
> | M1 ~ M2   | 0       |
> | M2 ~ M3   | 31      |
> | M3 ~ M4   | 12      |
> | M4 ~ M5   | 2       |
>
> Distribution of the magnitude of the 45 earthquakes in the evaluation set and the performance on DASFormer and Aggregation-0(STA/LTA baseline):
>
> |DASFormer          |         | AE   |      | EMD  |      | sliced EMD |      |
> |-----------|---------|------|------|------|------|------------|------|
> | Magnitude | Numbers | AUC  | F1   | AUC  | F1   | AUC        | F1   |
> | M0 ~ M1   | 0       | N/A  | N/A  | N/A  | N/A  | N/A        | N/A  |
> | M1 ~ M2   | 1       | N/A  | N/A  | N/A  | N/A  | N/A        | N/A  |
> | M2 ~ M3   | 27      | 0.84 | 0.4  | 0.88 | 0.46 | 0.86       | 0.42 |
> | M3 ~ M4   | 14      | 0.89 | 0.59 | 0.92 | 0.58 | 0.93       | 0.6  |
> | M4 ~ M5   | 3       | N/A  | N/A  | N/A  | N/A  | N/A        | N/A  |
>
> |Aggregation-0 |         | AE   |      | EMD  |      | sliced EMD |      |
> |-----------|---------|------|------|------|------|------------|------|
> | Magnitude | Numbers | AUC  | F1   | AUC  | F1   | AUC        | F1   |
> | M0 ~ M1   | 0       | N/A  | N/A  | N/A  | N/A  | N/A        | N/A  |
> | M1 ~ M2   | 1       | N/A  | N/A  | N/A  | N/A  | N/A        | N/A  |
> | M2 ~ M3   | 27      | 0.74 | 0.24 | 0.7  | 0.28 | 0.7        | 0.25 |
> | M3 ~ M4   | 14      | 0.73 | 0.24 | 0.69 | 0.26 | 0.7        | 0.25 |
> | M4 ~ M5   | 3       | N/A  | N/A  | N/A  | N/A  | N/A        | N/A  |
>
>
> For the earthquakes in the range of M2 ~ M3 and M3 ~ M4, we can see that the performance is becoming better for larger magnitude earthquakes. In each level, the performance of DASFormer is better than Aggregation-0 (STA/LTA baseline). For the earthquakes in the range of M0 ~ M2 and M4 ~ M5, the number of samples is too small to draw a conclusion.

---

### Official Review · Reviewer_8ntH · 2023-11-04

**Soundness:** 3 good
**Presentation:** 3 good
**Contribution:** 3 good
**Rating:** 5
**Confidence:** 5

**Summary:**

This article leverages a self-supervised pretrained network model to enhance seismic detection efforts, addressing the challenge of insufficient labeling for distributed acoustic sensing (DAS) data. The network architecture is thoughtfully designed with a Swin U-Net and Convolutional U-Net, allowing it to efficiently capture the spatio-temporal characteristics of DAS data. The network's performance is further enhanced through the implementation of various strategies, including convolutional pathing, DASFormer blocks, and noise injection. The primary contribution of this article lies in its introduction of a self-supervised pre-training framework for DAS seismic monitoring, even in cases where labeled data is unavailable. This framework is validated in downstream tasks such as earthquake detection and P/S seismic phase pickup.

**Strengths:**

1. The research problem is of great importance to society and is not well studied.
2. Using a self-supervised pre-trained network model to enhance seismic detection efforts sound promising to address the challenge of insufficient labeling for distributed acoustic sensing (DAS) data.
3. The network architecture is thoughtfully designed with a Swin U-Net and Convolutional U-Net.
4. This framework is validated in downstream tasks such as earthquake detection and P/S seismic phase pickup.

**Weaknesses:**

This article introduces a novel self-supervised pretrained model for DAS data, yet it falls short in demonstrating its superiority over existing methods. The reasons for this decision are outlined below:
1. While the paper compares different structures of pre-trained Benchmark models for seismic detection, these benchmarks suffer from poor representation and fairness issues:
(1)	The 'Aggregation-0' and 'Aggregation-inf' models provided in Table 1 lack clear descriptions and references to existing work, making it difficult to understand their characteristics and suitability (I am not sure if this is the LTA/STA method?)
(2)	The use of the j-DAS method, primarily designed for DAS data denoising, in a seismic detection comparison is considered unfair and may not yield equitable results.
(3)	The absence of an effective comparison with established DAS seismic detection methods, such as CNN-RNN (Hernández et al., 2022) and PhaseNet-DAS (Zhu et al., 2023), diminishes the paper's ability to demonstrate the effectiveness of its proposed approach
2. The resampling of the dataset from 2000-4000Hz to 10Hz for pre-training lacks a comparative verification of its impact on downstream tasks. Notably, the usual data sampling frequency for seismometer-based seismic detection tasks is within the range of 100-250Hz.
3. The paper does not adequately address the low signal-to-noise ratio (SNR) characteristic of DAS data, failing to provide clear information about the magnitude distribution and SNR of the training data. This omission leaves unanswered questions regarding the model's ability to detect earthquakes below a certain magnitude or signal-to-noise ratio.
4. The paper fails to make a clear distinction between P/S phase pickup and seismic detection tasks. For seismic monitoring purposes, distinguishing between P/S phases might be unnecessary, causing confusion in the results presented in Table 1. Additionally, the paper lacks a detailed description of the dataset and evaluation metrics for the downstream task of accurate seismic phase-on-arrival pickup. It only mentions fine-tuned training with 20 labeled data points in Figure 4, leaving crucial aspects of this task unexplained.
5. The paper lacks ablation experiments to elucidate the quantity of training data necessary for effective seismic monitoring.


The paper has many imprecise parts. Here are a few
1.  The 'Deep Learning on DAS Data' section of the RELATED WORKS lacks sufficient relevance to DAS data, apart from PhaseNet-DAS and j-DAS. More pertinent related work should be incorporated to provide a comprehensive context.
2. In the 'Time Series Modeling' section, the paper emphasizes the limitations of existing time series models in incorporating spatial information. However, there are network structures that can effectively integrate both spatial and temporal analyses. Including and comparing these structures with the work presented in this section would enhance the paper's precision.
3. The paper emphasizes the role of the coarse and fine steps in its approach, but it does not include ablation experiments to demonstrate the individual contributions and effectiveness of these two components. Such experiments would provide valuable insights into the significance of each step.
4. Figure 3 exclusively compares the earthquake detection effects of different prediction methods, with DASFormer being the only one shown to predict effective information. To enhance credibility, the paper should consider verifying and comparing these solutions with examples where other methods are equally capable of prediction, thus providing a more comprehensive assessment of their capabilities.

**Questions:**

please check the weakness part

---

> ### Author Response · Authors · 2023-11-23
>
> **Q1:**  The 'Aggregation-0' and 'Aggregation-inf’ lack description.
>
> **A1:** (1) "Aggregation-0" is a multi-variate version of STA/LTA method. It calculates the average of the absolute values of the STA/LTA scores across all variables, which is used as our anomaly score.
>
> (2) "Aggregation-inf" functions as a ransdom detector with no detection ability. It randomly assigns labels of 0 or 1 with a 50% probability for each sample.
>
> ---
>
> **Q2:** The use of the j-DAS method for seismic detection comparison is considered unfair and may not yield equitable results.
>
> **A2:** We understand the reviewer's concern that j-DAS seems more focused on denoising tasks. To the best of our knowledge, j-DAS is the only existing model that the authors claim to be a foundational model on DAS data. Therefore, considering the claimed potential capabilities of j-DAS, we decided to conduct comparison experiments with it.
>
> ---
>
> **Q3:** The paper does not effectively compare with established DAS seismic detection methods, diminishing its ability to demonstrate the effectiveness of its proposed approach.
>
> **A3:** Please refer to the Answer to A6.
>
> ---
>
> **Q4:** The resampling of the dataset from 2000-4000Hz to 10Hz for pre-training lacks a comparative verification of its impact on downstream tasks.
>
> **A4:** As you mentioned, the original sampling of our DAS data is 250Hz. Resampling the data from 250Hz to 10Hz allows us to retain the important features and characteristics of the seismic signals while reducing the data size and computational complexity. ****For seismic detection tasks, the lasting time of P waves typically spans a few seconds, while S waves can last even longer, up to several minutes. Given this context, a sampling frequency of 10Hz is sufficient to capture the necessary temporal information required for accurate seismic detection.
>
> ---
>
> **Q5:** The model's ability to detect earthquakes below a certain magnitude or signal-to-noise ratio.
>
> **A5:** Thanks for your suggestion. We provide magnitude distribution of our datasets here:
>
> We provide magnitude distribution of our datasets here:
>
> Distribution of the magnitude of the 45 earthquakes in the training set:
>
> | Magnitude | Numbers |
> |-----------|---------|
> | M0 ~ M1   | 0       |
> | M1 ~ M2   | 0       |
> | M2 ~ M3   | 31      |
> | M3 ~ M4   | 12      |
> | M4 ~ M5   | 2       |
>
> Distribution of the magnitude of the 45 earthquakes in the evaluation set and the performance:
>
> |DASFormer          |         | AE   |      | EMD  |      | sliced EMD |      |
> |-----------|---------|------|------|------|------|------------|------|
> | Magnitude | Numbers | AUC  | F1   | AUC  | F1   | AUC        | F1   |
> | M0 ~ M1   | 0       | N/A  | N/A  | N/A  | N/A  | N/A        | N/A  |
> | M1 ~ M2   | 1       | N/A  | N/A  | N/A  | N/A  | N/A        | N/A  |
> | M2 ~ M3   | 27      | 0.84 | 0.4  | 0.88 | 0.46 | 0.86       | 0.42 |
> | M3 ~ M4   | 14      | 0.89 | 0.59 | 0.92 | 0.58 | 0.93       | 0.6  |
> | M4 ~ M5   | 3       | N/A  | N/A  | N/A  | N/A  | N/A        | N/A  |
>
> |Aggregation-0 |         | AE   |      | EMD  |      | sliced EMD |      |
> |-----------|---------|------|------|------|------|------------|------|
> | Magnitude | Numbers | AUC  | F1   | AUC  | F1   | AUC        | F1   |
> | M0 ~ M1   | 0       | N/A  | N/A  | N/A  | N/A  | N/A        | N/A  |
> | M1 ~ M2   | 1       | N/A  | N/A  | N/A  | N/A  | N/A        | N/A  |
> | M2 ~ M3   | 27      | 0.74 | 0.24 | 0.7  | 0.28 | 0.7        | 0.25 |
> | M3 ~ M4   | 14      | 0.73 | 0.24 | 0.69 | 0.26 | 0.7        | 0.25 |
> | M4 ~ M5   | 3       | N/A  | N/A  | N/A  | N/A  | N/A        | N/A  |
>
>
> For the earthquakes in the range of M2 ~ M3 and M3 ~ M4, we can see that the performance is becoming better for larger magnitude earthquakes. In each level, the performance of DASFormer is better than Aggregation-0. For the earthquakes in the range of M0 ~ M2 and M4 ~ M5, the number of samples is too small to draw a conclusion.

---

> ### Author Response · Authors · 2023-11-23
>
> **Q6:** The paper fails to make a clear distinction between P/S phase pickup and seismic detection tasks, causing confusion in the results presented in Table 1.
>
> **A6:** P/S phase pickup refers to the identification and classification of seismic wave phases, specifically the arrival times of the P (primary) and S (secondary) waves, which focuses on accurately detecting and labeling these specific phases point-wise in DAS data.  On the other hand, seismic detection task (in Table 1) involves identifying and classifying anomalies or abnormal patterns in DAS data. The main differences between P/S phase pickup and seismic detection tasks can be summarized in the following aspects:
>
> (1) **Wave identification**: In P/S phase pickup, the method needs to accurately distinguish between P and S waves in seismic data. But seismic detection tasks do not require this specific wave identification, only with earthquake events.
>
> (2) **Data representation**: P/S phase pickup typically provides point-wise results, indicating the exact arrival times of P and S waves. In contrast, seismic detection tasks focus on determining whether there is an event or anomaly within a specific time span, without the need for precise point-wise results. Moreover, our training/valid/testing splits are different from PhaseNet/PhaseNet-DAS
>
> (3) **Learning setting**: P/S phase pickup methods are typically implemented in a supervised learning setting, where labeled data with ground-truth P/S wave information is required for training. However, our seismic detection task is addressed in an unsupervised learning setting like anomaly detection, as obtaining labeled data for anomalies can be challenging.
>
> ---
>
> **Q7:**: The paper lacks a detailed description of the dataset and evaluation metrics for the downstream task of accurate seismic phase-on-arrival pickup.
>
>
> **A7:**: We will update more description of the dataset, including the distribution of the magnitude of the earthquakes and the SNR analysis of the training data. Actually, we don't have evaluation metrics for the downstream task of accurate seismic phase-on-arrival pickup. We just visually show the results. It is worth mentioning that while resampling the frequency to 10Hz is sufficient for earthquake detection and monitoring, it may not be adequate for tasks like P/S phase pickup that require original-resolution (250Hz) data. We acknowledge that generalizability for original-resolution data is an area we plan to explore in the future.
>
> ---
>
> **Q8:** The paper lacks ablation experiments to elucidate the quantity of training data necessary for effective seismic monitoring.
>
> **A8:** We agree that ablation studies on the quantity of training data can investigate the performance especially the scalability of DASFormer. The results of this ablation study is shown in following table:
>
> |                      | AE   |      | EMD  |      | sliced EMD |      |
> |----------------------|------|------|------|------|------------|------|
> |                      | AUC  | F1   | AUC  | F1   | AUC        | F1   |
> | 5 earthquake events  | 0.71 | 0.34 | 0.74 | 0.37 | 0.72       | 0.37 |
> | 10 earthquake events | 0.78 | 0.37 | 0.80 | 0.44 | 0.80       | 0.42 |
> | 20 earthquake events | 0.86 | 0.52 | 0.88 | 0.51 | 0.88       | 0.54 |
> | 45 earthquake events (Full data) | 0.88 | 0.55 | 0.89 | 0.52 | 0.90       | 0.56 |
>
> ---
>
> **Q9:** The "Deep Learning on DAS Data" section of the RELATED WORKS lacks sufficient relevance to DAS data, apart from PhaseNet-DAS and j-DAS.
>
> **A9:** In the revision, we will make sure to include more relevant works that are related to DAS data. It would be better if you can give some suggestions on the related works. Thanks!

---

> ### Author Response · Authors · 2023-11-23
>
> **Q10:** The paper emphasizes the limitations of existing time series models in incorporating spatial information, but does not include network structures that can effectively integrate both spatial and temporal analyses.
>
>
> **A10:** Thanks for the suggestion. There are many existing spatio-temporal time series methods aim to capture and integrate spatial information for multi-variate time series data. We select three representative methods (Autoformer, GDN, and Crossformer) and compare them with DASFormer. The results are shown in Table 1 and Table 6. Autoformer and Crossformer are two Transformer-based methods utilizing spatial dependency in multi-variate time series. GDN is a graph-based method that uses a graph neural network to capture the spatial dependency of the multi-variate time series.
>
> There are many existing spatio-temporal time series methods aim to capture and integrate spatial information for multi-variate time series data. However, to the best of our knowledge, all of them are order-agnostic models (with GNNs or Transformers) to capture spatial dependency, which typically do not consider the order or sequence of the variates in the data. This can be problematic in the context of DAS data, where the order of variates is crucial as it represents the location of stations along the fiber optic cable.
>
> ---
>
> **Q11:** The paper emphasizes the role of the coarse and fine steps in its approach, but it does not include ablation experiments to demonstrate the individual contributions and effectiveness of these two components.
>
> **A11:** Thanks for the suggestion. We conduct ablation studies on the three steps (Coarse, Coarse-to-Fine, Coarse-to-Fine with Discriminator). The results are shown in the following table:
>
> |                       | AE   |      | EMD  |      | sliced EMD |      |
> |-----------------------|------|------|------|------|------------|------|
> |                       | AUC  | F1   | AUC  | F1   | AUC        | F1   |
> | Coarse-only           | 0.79 | 0.41 | 0.81 | 0.44 | 0.83       | 0.47 |
> | Coarse-to-Fine        | 0.85 | 0.52 | 0.87 | 0.53 | 0.87       | 0.50 |
> | Coarse-to-Fine w/ GAN | 0.88 | 0.55 | 0.89 | 0.52 | 0.90       | 0.56 |
>
> The results demonstrate the effectiveness of the coarse-to-fine framework and the discriminator. We will add these results to the revision.
>
> ---
>
> **Q12:** Figure 3 exclusively compares the earthquake detection effects of different prediction methods, with DASFormer being the only one shown to predict effective information, lacking a comprehensive assessment of the capabilities of different methods.
>
>
> **A12:**  In fact, all samples of baselines are like "not capable of prediction", we didn't select the worse ones. We suggest to consider the results of both aggregation-0 (sta/lta) and aggregation-inf in Table 1 together with Figure 3. Note that if an algorithm consistently predicts 0 as the predicted future DAS signals, which corresponds to a completely white region in the forecasting region of Figure 3, it can be seen as a special case of the sta/lta algorithm.  This means that the expected long-term average is 0, and the difference between the real value and 0 (LTA) is kind of equivalent to the sta/lta value (assuming the expected average of lta remains constant). Given this context, we can observe that almost all methods (except FEDformer) perform similarly to aggregation-0 (sta/lta) in Table 1. And in Figure 3, we can see that most of the baselines (except FEDformer) output values close to white (close to 0 in values) , resulting in similar results to aggregation-0.

---

### Author Response · Authors · 2023-11-23
**Overall comment**

Dear Reviewers,

We appreciate the reviewers for their careful reading and constructive remarks.

To summarise the initial reviews, we thank all reviewers for agreeing that our work fills a gap in a new research domain. However, the reviewers also raised some general concerns.

**General Concern 1** The distribution of the magnitude of the earthquakes and the performance for each magnitude level with different SNR ratios.

**A:** We counted the distribution of the data set, supplemented relevant experiments, and compared the results of the baseline and our method at different magnitudes. Please refer to the individual answers to the reviewers.

**General Concern 2** Concerning ablation studies on the quantity of training data and three steps (Coarse, Coarse-to-Fine, Coarse-to-Fine with Discriminator).

**A:** We supplemented such experiments below and will include them in the revision. Please refer to the individual answers to the reviewers.

**General Concern 3** Comparison with other DAS earthquake detectors like PhaseNet-DAS.

**A:** We explained the difference between P/S phase pickup and seismic detection tasks and why we did not compare with PhaseNet-DAS in the individual answers to the reviewers.

**General Concern 4** The performance of baselines and the meaning of Aggregation-0.

**A:** We explained that Aggregation-0 is a multi-variate version of STA/LTA method. The similar performance of baselines and Aggregation-0 is not because the baselines are not capable of prediction but because they perform like the STA/LTA method. We explained the reason in the individual answers to the reviewers.


We reply to specific questions individually below.

---

### Meta-Review · Area_Chair_1772 · 2023-12-05

**Metareview:**

This paper presents a self-supervised algorithm for learning from Earthquake patterns.

After the rebuttal and AC-reviewer discussion stage, the final scores of this paper are 5/5/5/6. One reviewer changed his/her score from 6 to 5. The AC asked for discussion but nobody responded. The AC looked into the case by reading the paper and the rebuttal. The good point lies in the studied problem which is important to society, but the paper lacks novelty in the AI community -- the setting and algorithm are similar to the defect detection problem in the CV community. The reviewers also pointed out other issues such as a lack of technical details and ablations. Finally, the AC chose to concur with the majority of reviewers to recommend rejection.

**Justification For Why Not Higher Score:**

The average score falls below the acceptance threshold.

**Justification For Why Not Lower Score:**

N/A

---

### Decision · Program_Chairs · 2024-01-16

Reject